# Beyond Search: Direct Model Guidance for Steerable Synthesis Planning

## Abstract

Synthetic chemists often need to incorporate domain-specific constraints and preferences into synthesis plans, such as preferred reaction types or available starting materials, motivating the development of steerable synthesis planning methods. While previous approaches navigate the search space with a frozen single-step model for plausible multi-step routes, we demonstrate that directly guiding the single-step retrosynthesis model enables exploration of previously inaccessible chemical spaces during generation. Specifically, we invoke guidance to modify the logits of an autoregressive seq2seq retrosynthesis model, enabling conditioning on various properties without retraining. Empirically, we demonstrate that while commonly used single-step models struggle to find routes with chemically feasible single-step reactions throughout the entire synthesis plan, our method generates synthesis routes of equal or better quality than template-based approaches while satisfying the specified constraints.

## 1 Introduction

Computer-aided synthesis planning has emerged as a critical tool for accelerating drug discovery and materials development, yet real-world synthetic chemistry demands more than just feasible routes. Chemists require routes that align with their preferred reaction methodologies (Nicolaou & Sorensen, 1996), available materials (Yu et al., 2024), and process constraints (Griffin et al., 2023). These domain-specific requirements motivate the need for steerable synthesis planning methods that can incorporate chemist preferences and constraints directly into route generation. Current approaches to steerable synthesis planning predominantly operate at the search level, using frozen single-step retrosynthesis models to generate fixed sets of reaction suggestions that are then filtered (Segler et al., 2018), reranked (Lin et al., 2022), or guided through modified search heuristics (Armstrong et al., 2024) (see fig. 1 for a comparison).

Fixing the single-step model, however, limits the exploration of the chemical space available to the search algorithm. We propose adapting classifier guidance, a technique established in controllable text generation, to neural retrosynthesis. While classifier guidance has been successfully applied to natural language tasks, its application to chemical synthesis planning presents unique challenges: the discrete, structured nature of molecular representations, the need for chemically valid outputs, and the importance of maintaining reaction feasibility. Our method modifies the logits of a pre-trained single-step retrosynthesis model to bias generation toward desired properties, such as specific reaction types or compatibility with available starting materials, without requiring model retraining. Critically, we provide the first theoretical analysis demonstrating that token-level guidance can provably access regions of chemical space unreachable by unguided beam search, even with arbitrarily large beam widths, a guarantee previously unexplored in either text generation or chemical synthesis contexts.

Our approach offers several key advantages. It requires no model retraining, guidance strength can be tuned at inference time, and multiple property constraints can theoretically be composed through classifier combination. We demonstrate our method using two properties of broad interest: reaction type control and starting material-constrained synthesis, though the framework extends to any measurable reaction property including yield prediction, toxicity minimization, or cost optimization.

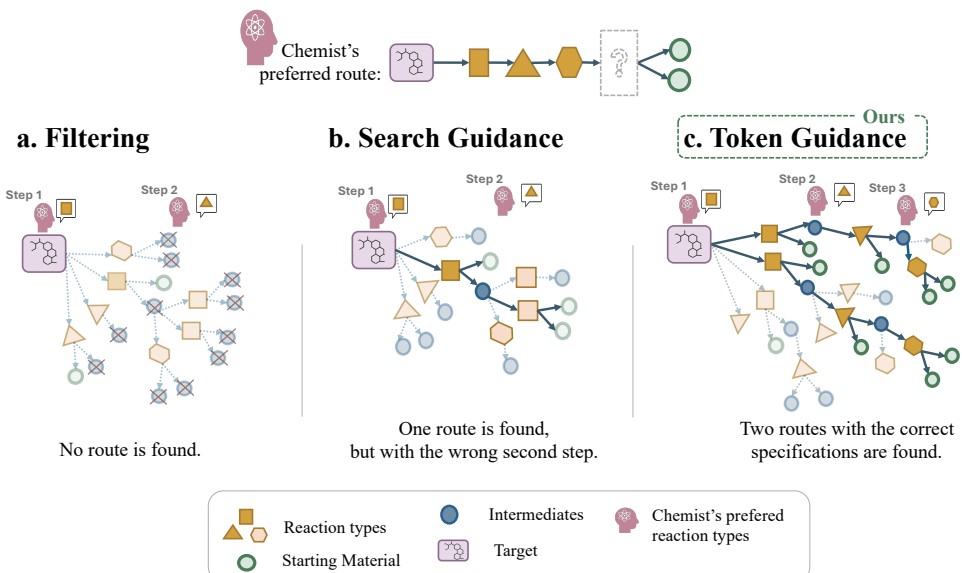

Figure 1: Comparison of approaches to steerable synthesis planning. Filtering (a) drops nodes that do not satisfy the required criteria (e.g., a specific reaction type) from the search tree. Search-based guiding (b) relies on biasing the value function of the search algorithm to favor nodes meeting the given criteria, but it can expand other nodes too if no such node is found (e.g. at the second expansion stage, nodes from a square reaction are expanded because no triangle reactions are available). Our method, direct guidance, provides the desired criteria as input to the single-step model, and can thus produce more precursors of the right reaction type (as shown in the first reaction step) and suggest precursors unlikely to happen otherwise (as shown with the triangle type in the second step).

## 2 PRELIMINARIES

### 2.1 SYNTHESIS PLANNING

Synthesis planning is the process of defining a step-by-step route to produce a target molecule from a set of starting materials. While complete plans include things like reaction conditions and protocols, data-driven approaches typically focus on predicting only the precursors (molecules) involved in the plan. Formally, we define $\mathcal{M}$ as the space of all valid molecules, $\mathcal{R}$ as the space of all valid reactions, and $R = (\mathbf{r}, p)$ as a reaction made of reactants $\mathbf{r} \subset \mathcal{M}$ and a single main product $p \in \mathcal{M}$. Given a target molecule $p^* \in \mathcal{M}$ and a set of building blocks $B \subset \mathcal{M}$, a valid synthetic route is a set of reactions $S = \{(\mathbf{r}_1, p_1), \ldots, (\mathbf{r}_n, p_n)\}$ forming a directed acyclic graph (DAG) and satisfying two key constraints:

1. Each reactant is either a building block or a product of a previous reaction: $\forall i : m \in \mathbf{r}_i, m \notin B \Rightarrow \exists j$, s.t. $m = p_j$
2. The target molecule is the final product and not an intermediate: $\exists j$ s.t. $p^* = p_j, \forall i : p^* \notin \mathbf{r}_i$

When solving starting material-constrained synthesis planning, we add the constraint that a specified starting material $s^* \in \mathcal{M}$ must appear as a reactant (but not as a product) in at least one reaction:

3. Starting-material constrained synthesis: $\exists i$ s.t. $s^* \in \mathbf{r}_i, \forall j : s^* \neq p_j$

Data-driven synthesis planning typically decomposes the multi-step problem into single-step retrosynthesis predictions, where models predict plausible precursors for a given product molecule, which are then used as a policy by a search algorithm. Formally, we can represent any template-free single-step retrosynthesis (i.e. a prediction method that does not rely on pre-defined chemical templates) model as learning a conditional probability distribution $p_\theta(\mathbf{r} \mid p)$, where $\mathbf{r}$ represents reactants and $p$ the product. One of the best template-free methods is Root-aligned SMILES

(`R-SMILES`) (Zhong et al., 2022), which aligns molecular representations to minimize the edit distance between products and reactants, enabling more effective learning of chemical transformations. Given its success, this work focuses on expanding `R-SMILES` to guided generation.

## 2.2 CLASSIFIER GUIDANCE FOR AUTOREGRESSIVE MODELS

Classifier guidance (Dhariwal & Nichol, 2021b) is an established technique in generative models that allows steering generation toward desired properties during inference without retraining. Given property predicates $\{\phi_\alpha\}_{\alpha=1}^n$ where each $\phi_\alpha : \mathcal{R} \rightarrow \{0, 1\}$ evaluates whether a reaction satisfies property $\alpha$,[1] and a property predictor $p_\alpha(\phi_\alpha = 1 \mid \mathbf{r}, p)$, we can define a guided distribution using Bayes' rule:

$$p_\theta(\mathbf{r} \mid p, \phi_\alpha = 1) \propto p_\alpha(\phi_\alpha = 1 \mid \mathbf{r}, p) \cdot p_\theta(\mathbf{r} \mid p), \tag{1}$$

where $p_\theta(\mathbf{r} \mid p)$ is the base generative model. The guidance strength can be controlled via a temperature parameter $\lambda$:

$$p_\theta(\mathbf{r} \mid p, \phi_\alpha = 1) \propto \left[ p_\alpha(\phi_\alpha = 1 \mid \mathbf{r}, p) \right]^\lambda \cdot p_\theta(\mathbf{r} \mid p). \tag{2}$$

For autoregressive models that generate sequences token-by-token, this guidance is applied at the token level. Given a sequence $\mathbf{r} = (r_1, \ldots, r_L)$ where $r_k \in \mathcal{V}$ is the $k$-th token and $\mathbf{r}_{<k} = (r_1, \ldots, r_{k-1})$ is the partial sequence, the guided token distribution is:

$$p_\theta^\lambda(r_k \mid \mathbf{r}_{<k}, p, \phi_\alpha = 1) \propto \left[ p_\alpha(\phi_\alpha = 1 \mid \mathbf{r}_{\leq k}, p) \right]^\lambda \cdot p_\theta(r_k \mid \mathbf{r}_{<k}, p) \tag{3}$$

where $\mathbf{r}_{\leq k} = (r_1, \ldots, r_k)$ includes the current token. In practice, we operate in log-space:

$$\log p_\theta^\lambda(r_k \mid \mathbf{r}_{<k}, p, \phi_\alpha = 1) \propto \lambda \log p_\alpha(\phi_\alpha = 1 \mid \mathbf{r}_{\leq k}, p) + \log p_\theta(r_k \mid \mathbf{r}_{<k}, p). \tag{4}$$

Note that the classifier $p_\alpha(\phi_\alpha = 1 \mid \mathbf{r}_{\leq k}, p)$ must predict the property value for the complete reaction based on a partially completed sequence. When the vocabulary size is large, efficiency can be improved by evaluating only the top-$N$ most probable tokens under the base model (Yang & Klein, 2021).

## 3 METHOD: TOKEN-GUIDED SYNTHESIS PLANNING

### 3.1 PROBLEM FORMULATION

In token-guided synthesis planning, our goal is to generate routes where more reactions satisfy a desired property $\alpha$ (e.g., a specific reaction type) compared to unguided generation. Existing approaches to steerable synthesis planning operate at the search level. Methods like Neural Sym (Segler et al., 2018) filter single-step predictions to retain only property-satisfying reactions, while approaches like TangoStar (Armstrong et al., 2024) modify search heuristics to favor such reactions during route construction. However, these methods are fundamentally limited to the chemical space accessible by their underlying single-step model. When a desired property $\phi_\alpha$ is rare in the training data $\mathcal{D} = \{(\mathbf{r}_i, p_i)\}_{i=1}^N$ (i.e., $\mathbb{P}_{(\mathbf{r},p)\sim\mathcal{D}}[\phi_\alpha(\mathbf{r}, p) = 1] \ll 1$), the base retrosynthesis model $p_\theta(\mathbf{r} \mid p)$ trained via maximum likelihood estimation concentrates probability mass primarily in regions where $\phi_\alpha$ rarely holds, making such precursors inaccessible to the synthesis planner even with arbitrarily sophisticated search algorithms.

We propose a fundamentally different approach: directly guiding the token-level generation process of the single-step retrosynthetic model. By applying classifier guidance during autoregressive decoding, we modify the base model's token distributions to redirect probability mass toward underexplored regions of chemical space containing property-satisfying reactions, which we illustrate in fig. 2 for the synthesis of Acetophenone. This intervention at the generation level enables access to precursors that would be entirely filtered out by beam search using the unguided model. Mathematically, our objective is to maximize the expected property satisfaction for each single-step prediction:

$$\max_{\mathbf{r}} \mathbb{E}_{p_\theta^\lambda(\mathbf{r}|p,\phi_\alpha=1)} \left[ \mathbb{I}[\phi_\alpha(\mathbf{r}, p) = 1] \right] \tag{5}$$

---

[1]When dealing with continuous properties, such as yield, we can use a threshold to convert them to binary properties.

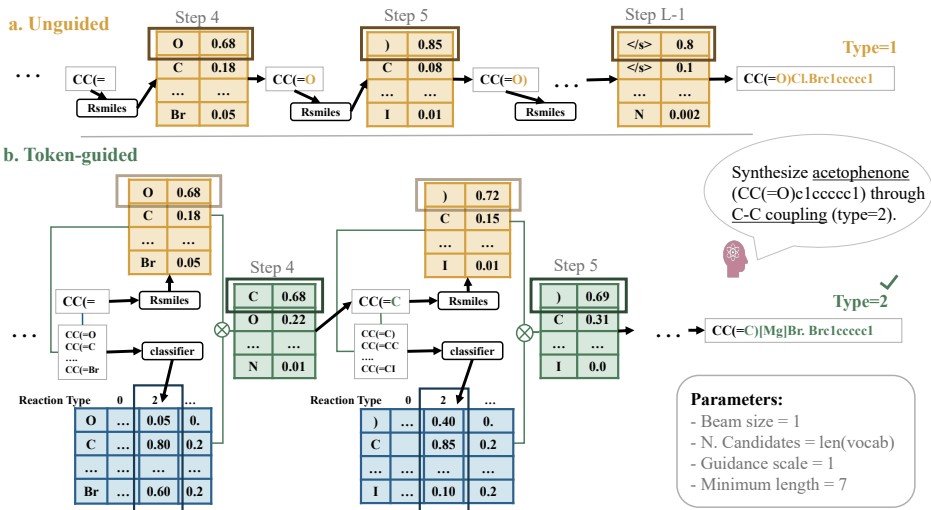

Figure 2: Illustrating token-level reranking using classifier-guidance in beam search. Figure (a) shows regular beam search starting from the 4th generation step. At step 4, because `C` is unlikely under the generator's distribution, the model is unable to generate precursor `CC(=C)[Mg]Br`, which belongs to the right reaction type (type 2). Figure (b) shows that using the classifier's probabilities, we can force the model to rerank the tokens, thus pushing `C` inside the beam.

where $p_\theta^\lambda(\mathbf{r} \mid p, \phi_\alpha = 1)$ is the guided distribution defined in section 2.2. By biasing each single-step generation toward the desired property, we expect the resulting multi-step routes to satisfy the property more frequently, without directly optimizing the route-level criterion.

## 3.2 TOKEN-LEVEL GUIDANCE FOR RETROSYNTHESIS

We implement classifier-guided generation for single-step retrosynthesis by modifying the beam search decoding of pre-trained seq2seq models. Our approach builds on Root-aligned SMILES (R-SMILES) (Zhong et al., 2022), one of the strongest template-free transformer-based retrosynthetic models. R-SMILES' key innovation is aligning molecular representations through canonicalization with different atom roots, which minimizes edit distance between products and reactants and enables more effective learning of chemical transformations.

Our method integrates classifier guidance by keeping the pre-trained R-SMILES weights frozen and combining its token-level predictions with those of a property predictor during beam search, as formalized in algorithm 1. The minimum length threshold $l_{\min}$ ensures the classifier has sufficient context to make reliable predictions, while the guidance scale $\lambda$ controls the trade-off between base model likelihood and property satisfaction.

The property classifiers $p_\alpha(\phi = 1 \mid \mathbf{r}_{\leq k}, p)$ must predict whether a complete reaction satisfies property $\phi$ based on a partially generated reactant sequence $\mathbf{r}_{\leq k}$ and product $p$. We implement these classifiers as Transformers following the same architectural design and data augmentation strategy as R-SMILES, but with a property prediction head and smaller network capacity. Training involves generating synthetic partial sequences by randomly truncating reactant SMILES from the training data at various completion ratios, enabling the classifier to learn prediction from incomplete inputs. Full architectural details and training procedures are provided in section F. The specific properties we explore, namely reaction type control and starting material constraints, and their practical relevance are discussed in section 3.4.

The key insight enabling our approach is that classifier feedback reranks tokens at each generation step, allowing sequences with low base model probability but high property probability to enter the beam. This token-level intervention provides access to regions of chemical space that unguided beam search cannot reach, as we formalize in the following section.

---

**Algorithm 1** Token-Guided Beam Search (Single Generation Step $t$)

---

**Require:** Current beam $\mathcal{B}$, product $p$, guidance scale $\lambda$, min. length $l_{min}$, step $t$, top-N $N$
**Ensure:** Updated beam $\mathcal{B}'$
1: $\mathcal{C} \leftarrow \emptyset$
2: **for** each sequence $s \in \mathcal{B}$ **do**
3:     Get base logits: $\text{logits}_\theta \leftarrow \log p_\theta(r_t|s, p)$
4:     **if** $t \geq l_{min}$ **then**
5:         $\mathcal{T} \leftarrow \text{TopN}(\text{logits}_\theta, N)$ {Select top-N tokens}
6:         **for** each $r_t \in \mathcal{T}$ **do**
7:             $s' \leftarrow s \oplus r_t$ {Concatenate token}
8:             $c \leftarrow \log p_\alpha(\phi = 1|s', p)$ {Classifier score}
9:             $\text{logits}_\theta[r_t] \leftarrow \text{logits}_\theta[r_t] + \lambda \cdot c$
10:        **end for**
11:        Renormalize: $\text{logits}_\theta \leftarrow \text{logits}_\theta - \text{LogSumExp}(\text{logits}_\theta)$
12:     **end if**
13:     **for** each $r_t \in V$ **do**
14:         Add $(s \oplus r_t, \text{logits}_\theta[r_t])$ to $\mathcal{C}$
15:     **end for**
16: **end for**
17: $\mathcal{B}' \leftarrow \text{TopK}(\mathcal{C}, k)$
18: **return** $\mathcal{B}'$

---

### 3.3 THEORETICAL ANALYSIS

While classifier guidance has been applied to autoregressive text generation (Yang & Klein, 2021), prior work uses greedy decoding or sampling-based methods without formal analysis of reachability guarantees. Our work provides the first theoretical framework demonstrating that classifier-guided beam search can provably access regions of the output space that remain unreachable to the base model, even with arbitrarily large beam widths. This analysis is particularly important for structured generation tasks like chemical synthesis where beam search is the standard decoding strategy and coverage guarantees directly impact practical utility. The key insight is that classifier feedback reranks tokens at each generation step based on both base model likelihood and property probability. This reranking can elevate sequences with low base model probability $p_\theta(\mathbf{r} \mid p)$ but high property probability $p_\alpha(\phi = 1 \mid \mathbf{r}, p)$ above sequences with high base model probability but low property probability, enabling access to previously unreachable precursors.

**Theorem 1** (Reaching New Regions of Chemical Space). *Let $\beta_k^{(l)}$ be the score of the $k$-th beam (i.e., the beam with the lowest accepted score) at generation step $l$, and $R_\phi = \{r \in \mathcal{V}^L \mid \phi_\alpha(r, p) = 1, p_\theta(r \mid p) < \beta_k^{(l)}\}$ be the set of all property-satisfying sequences with base model probability less than $\beta_k^{(l)}$. Assume $R_\phi \neq \emptyset$ and define $r^* = \arg\max_{r \in R_\phi} p_\theta(r \mid p)$ as the property-satisfying sequence with the highest base model probability among those excluded by unguided beam search. Assume the classifier $p_\alpha$ satisfies for all $l \in [1, L]$:*

$$p_\alpha(\phi_\alpha = 1 \mid r^*_{\leq l}, p) \geq c_1 \tag{6}$$

$$p_\alpha(\phi_\alpha = 1 \mid r_{\leq l}, p) \leq c_2, \quad \text{when } \phi_\alpha(r, p) = 0 \tag{7}$$

*where $0 < c_2 < c_1 \leq 1$. Then:*

1. *Unguided beam search ($\lambda = 0$) excludes $r^*$ from the final output,*

2. *There exists $\lambda^* > 0$ such that guided beam search with guidance scale $\lambda \geq \lambda^*$ includes $r^*$ in the final output.*

This theorem guarantees that with sufficient guidance strength, our method can recover property-satisfying sequences that unguided beam search misses entirely. The classifier condition in eq. (8) requires the property predictor to be sufficiently accurate: it must assign high probability to property-satisfying sequences and low probability to non-property-satisfying sequences. This condition is easier to satisfy when the classifier has more context, motivating delayed guidance.

**Corollary 3.1** (Delayed Guidance). *Under the conditions of theorem 2, suppose the classifier satisfies the error bounds in eq.* (8) *only for partial sequences with $l \geq l_{\min}$. If $r^*_{\leq l_{\min}}$ remains in the beam during unguided generation (i.e., $\log p_\theta(r^*_{\leq l_{\min}} \mid p) \geq \beta_k^{(l_{\min})}$), then guided beam search with guidance applied at $l \geq l_{\min}$ includes $r^*$ in its final output for sufficiently large $\lambda$.*

Corollary A.1 formalizes the effectiveness of delayed guidance: if the target sequence remains in the beam until sufficient context accumulates for reliable classifier predictions, guidance can successfully steer generation toward it. This motivates our use of the minimum length threshold $l_{\min}$ in algorithm 1, ensuring guidance is applied only when the classifier has adequate input to distinguish property-satisfying from non-property-satisfying completions. Complete proofs and additional discussion are provided in section A.1.

### 3.4 RELEVANT APPLICATIONS

We focus on two properties of practical relevance: reaction type control and starting material-constrained synthesis. We note that our framework naturally extends to any reaction property with data available for training classifiers, and implementation details are discussed in section C.

Reaction types (or classes) represent high-level groupings of reactions based on their underlying mechanisms, as formalized by Carey's classification system (Carey et al., 2006). These categories capture the structural transformations that help reason about chemical reactions. Controlling reaction mechanisms allows chemists to explore alternative synthetic strategies—for instance, testing whether a target accessible via oxidation could also be reached through a coupling approach that better suits their laboratory capabilities.

In multi-step planning, a chemist can provide a series of desired reaction types and our framework can help find routes that satisfy these constraints, as shown in fig. 1. In starting-material-constrained synthesis planning, chemists seek to design synthesis routes that specifically incorporate target starting materials such as renewable feedstocks or waste products, to valorize underutilized chemical resources, reduce waste, and address availability constraints (Wołos et al., 2022). Similar multi-step reasoning applies: chemists can specify key starting materials and our framework can discover whether viable routes incorporating them exist. We guide the retrosynthesis model to generate precursors with high structural similarity to the desired starting material, as measured by the Tanimoto coefficient, a structural similarity metric ranging from 0 to 1 based on shared molecular features (formula in section C).

## 4 EXPERIMENTS

The goal of the experiment section is to 1) show how classifier-guidance leads to exploring a different chemical space (through single-step retrosynthesis), and 2) show how guiding at the single-setup level leads to improving the quality of the multi-step synthesis routes (through multi-step synthesis planning).

### 4.1 SINGLE-STEP EVALUATION

We evaluate token-level guidance on single-step retrosynthesis to assess whether it: (1) explores different chemical spaces than unguided generation, (2) increases property satisfaction without compromising sample validity, and (3) improves ground truth recovery rates.

**Experimental setup** We evaluate on USPTO-190-Steps (individual reactions from USPTO-190 test routes (Chen et al., 2020); processing details in section B). Using Syntheseus (Maziarz et al., 2023), we compare against all available baselines plus NeuralSym (Segler & Waller, 2017), a template-based method commonly used in multi-step planning (Chen et al., 2020; Yu et al., 2024). All methods generate 100 samples per product to ensure equal computational budget, which are then filtered by reaction type or tanimoto similarity with a threshold of 0.2, depending on the guidance target. For guided generation, we test 3 guidance scales per product for both reaction type and tanimoto similarity: $\lambda \in \{0.5, 1.0, 1.5\}$ for reaction type, and $\lambda \in \{5, 10, 20\}$ for tanimoto similarity, with a minimum length of 15. To stay within the computational budget of 100 samples

per product while using the right parameter per product, we first generate 10 samples per parameter, then choose the best parameter per product, and generate the remaining 70 samples with the best parameter. This way we guarantee an ideal parameter per product and maintain fairness to other baselines. More details on this procedure are provided in **??**. For models trained on USPTO-50k, we include the following metrics: top-k accuracy (fraction of products with ground truth in top-k filtered samples), unique samples (diversity), average round-trip accuracy per product, average class accuracy per product, and average tanimoto similarity to starting material per product.

Table 1: Single-step retrosynthesis evaluation on USPTO-190-Steps. Token-guided R-SMILES achieves higher property satisfaction (correct reaction class: $17-23\%$ vs $2-3\%$ unguided; Tanimoto similarity to starting material: $0.34-0.35$ vs $0.32$) while maintaining competitive top-k accuracy and sample diversity. All methods generate 600 samples per product with equal computational budget.

| Method | Temp. | Exact route (↑) | top-1 (↑) | top-5 (↑) | top-50 (↑) | Unique samples (↑) | Correct class (↑) | TA to SM (↑) | Correct R.Trip (↑) |
|---|---|---|---|---|---|---|---|---|---|
| *Trained on USPTO-50k* | | | | | | | | | |
| Localr<etro | ✓ | **0.04** | 0.23 | 0.43 | 0.52 | 0.24 | 0.06 | 0.32 | 0.11 |
| RetroKNN | ✓ | **0.04** | 0.26 | 0.43 | 0.51 | 0.25 | 0.06 | 0.32 | 0.11 |
| MHNReact | ✓ | 0.02 | 0.22 | 0.38 | 0.48 | 0.09 | 0.02 | **0.33** | 0.04 |
| Megan | × | 0.02 | 0.22 | 0.38 | 0.49 | 0.33 | 0.10 | 0.32 | 0.12 |
| Graph2Edits | × | 0.01 | 0.26 | 0.41 | 0.49 | 0.15 | 0.04 | 0.30 | 0.02 |
| Chemformer | × | 0.02 | 0.27 | 0.38 | 0.44 | 0.04 | 0.01 | 0.31 | 0.01 |
| Rsmiles | × | 0.03 | 0.27 | 0.44 | 0.51 | 0.06 | 0.02 | 0.32 | 0.02 |
| Rsmiles-TG$_{rxn}$ | × | **0.04** | **0.31** | **0.46** | **0.55** | **0.49** | **0.17** | 0.32 | **0.15** |
| *Trained on USPTO-190* | | | | | | | | | |
| NeuralSym | ✓ | 0.16 | 0.36 | 0.57 | 0.69 | 0.04 | 0.01 | 0.33 | 0.02 |
| Rsmiles | × | 0.28 | 0.46 | 0.68 | 0.80 | 0.10 | 0.03 | 0.34 | 0.02 |
| Rsmiles-TG$_{rxn}$ | × | 0.31 | **0.48** | **0.70** | **0.81** | 0.67 | **0.23** | 0.34 | **0.11** |
| Rsmiles-TG$_{sim}$ | × | **0.42** | 0.45 | 0.66 | 0.79 | **0.68** | 0.22 | **0.35** | **0.11** |

**Results on USPTO-190-Steps** Table 8 shows that token-guided R-SMILES achieves $31\%$ exact route recovery on USPTO-190-Steps, compared to $28\%$ for unguided R-SMILES and $16\%$ for NeuralSym, despite all methods generating 600 samples per product. Guidance toward specific starting materials proves particularly effective, recovering $42\%$ of routes versus $31\%$ for reaction type guidance and $14\%$ more than unguided reaction type guidance. Beyond exact matches, guided generation produces substantially higher-quality samples: $23\%$ match the target reaction class compared to $3\%$ for unguided R-SMILES, while maintaining $67\%$ unique samples per product. When compared to other models trained on USPTO-50k, the token-guided model matches template-based methods in exact route recovery while generating significantly more samples with correct reaction classes ($17\%$ vs $10\%$ for MEGAN). Critically, guided samples exhibit only $39\%$ overlap (Jaccard similarity) with unguided samples (fig. 3), demonstrating exploration of genuinely different chemical space rather than mere reranking. Among the 124 products where unguided R-SMILES fails to recover ground truth, guidance recovers $55\%$ via round-trip accuracy and $10\%$ via exact match (fig. 3), providing direct evidence that token-level guidance enables access to precursors inaccessible to the base model.

## 4.2 SYNTHESIS PLANNING EVALUATION

Next we explore the effect of token-guidance when combined with a search algorithm. The goal is to see whether guiding at the single-step level does indeed translate to better routes, including recovering more ground truth routes.

**Experimental setup** The search algorithm we use is Retro* (Chen et al., 2020). For each product, we generate 100 samples, and set the search limits to 500 iterations, 100 model calls, and 600 seconds per model call. We follow the same parameter selection procedure as for the single-step evaluation. For each expanded product in the tree, we first generate 10 samples with 3 possible parameters, choose the best parameter, and then generate the remaining 70 samples with the best

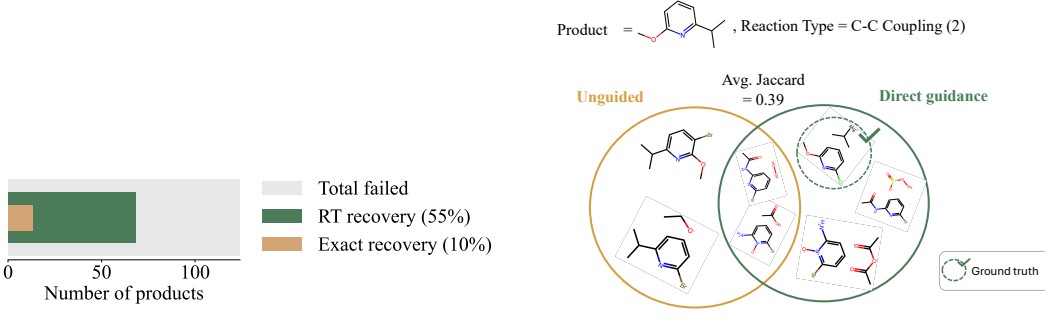

(a) Recovery rates for 124 products where unguided Rsmiles fail to recover the ground truth samples from USPTO-190

(b) Samples with and without guidance

Figure 3: Effect of guidance on Rsmiles samples. Guidance recovers 55% of failed products via round-trip accuracy and 10% via exact match, while increasing sample diversity and reaction type accuracy, in addition to only overlapping with 39% of the samples generated without guidance.

Table 2: Multi-step Synthesis Planning Search Metrics on USPTO-190. The table compares the performance of various retrosynthesis planning methods on the USPTO-190 dataset, encompassing unguided baselines, filtering (F), search-level guidance (G), and token-level guidance (T) tailored for reaction type ($_{rxn}$) and starting material similarity ($_{sim}$). Metrics include the overall solve rate and the solve rate when specifically satisfying starting material constraints, along with computational costs (model calls, total time), and the diversity of non-overlapping routes generated.

| Method | Temp. | Solve rate ($\uparrow$) | Solve rate with SM ($\uparrow$) | Model calls ($\downarrow$) | Time taken ($\downarrow$) | Non-overlapping routes per target ($\uparrow$) |
|---|---|---|---|---|---|---|
| *NeuralSym* | | | | | | |
| NeuralSym-unguided | $\checkmark$ | 0.47 | 0.16 | 87.96 | **28.44** | 0.83 |
| NeuralSym-F$_{rxn}$ | $\checkmark$ | 0.25 | 0.20 | 38.96 | 303.88 | 0.29 |
| NeuralSym-G$_{rxn}$ | $\checkmark$ | 0.43 | 0.26 | 63.96 | 589.83 | 0.65 |
| *Rsmiles* | | | | | | |
| Rsmiles-F$_{rxn}$ | $\times$ | 0.62 | 0.38 | 8.29 | 616.52 | 2.22 |
| Rsmiles-F$_{sim}$ | $\times$ | 0.20 | 0.18 | **1.99** | 148.24 | 0.83 |
| Rsmiles-G$_{rxn}$ | $\times$ | 0.81 | 0.50 | 8.15 | 635.81 | 5.27 |
| Rsmiles-G$_{sim}$ | $\times$ | 0.86 | 0.64 | 8.34 | 635.46 | 5.07 |
| *Rsmiles-T* | | | | | | |
| Rsmiles-T$_{rxn}$ | $\times$ | **0.92** | 0.47 | 7.34 | 641.99 | 5.92 |
| Rsmiles-T$_{sim}$ | $\times$ | 0.86 | 0.52 | 6.98 | 648.54 | **7.43** |
| Rsmiles-TG$_{rxn}$ | $\times$ | 0.88 | 0.53 | 8.52 | 634.77 | 5.73 |
| Rsmiles-TG$_{sim}$ | $\times$ | 0.90 | **0.66** | 6.84 | 642.21 | 6.21 |

parameter. Other baselines are given a similar chance through the filtering set-up. We evaluate both the search metrics (solve rate, solve rate with starting material, number of model calls, time taken, and number of non-overlapping routes), as well as metrics to evaluate the quality of the routes recovered, which, to the best of our knowledge, has never been done before, including exact route recovery, average reaction type accuracy, reaction name identification, and round-trip accuracy across all generated routes per product.

**Results on USPTO-190** Token-guided R-SMILES achieves a 92% solve rate on USPTO-190 when guiding toward specific reaction types, compared to 81% for unguided R-SMILES and 47% for NeuralSym (table 2). For starting material-constrained synthesis, our method achieves 90% solve rate with 66% of routes successfully incorporating the target starting material, substantially outperforming both search-level guidance (64%) and filtering approaches (38%). Notably, token guidance generates 6.2 non-overlapping routes per target on average, nearly three times more diverse solutions than search-level guidance (table 2). The quality of generated routes also improves significantly: 49% of reactions match the specified reaction type compared to 23% for unguided

Table 3: Quality of Multi-step Synthesis Routes Generated by Various Methods on USPTO-190. This table evaluates the quality of synthesis routes produced by different planning methods (unguided, filtering (F), search-level guidance (G), and token-level guidance (T) for reaction type ($_{rxn}$) and starting material similarity ($_{sim}$)) on the USPTO-190 dataset. Key metrics include exact route recovery, round trip plausibility, adherence to specified reaction types (Reaction type, Reaction name), and the percentage of routes successfully incorporating the desired starting material.

| Method | Temp. | Exact route ($\uparrow$) | Round trip route ($\uparrow$) | Reaction type ($\uparrow$) | Reaction name ($\uparrow$) | Perc. of routes with SM per target ($\uparrow$) |
|---|---|---|---|---|---|---|
| *NeuralSym* | | | | | | |
| NeuralSym-unguided | $\checkmark$ | 0.01 | 0.11 | 0.02 | 0.04 | 0.13 |
| NeuralSym-F$_{rxn}$ | $\checkmark$ | 0.09 | 0.16 | 0.17 | 0.02 | 0.14 |
| NeuralSym-G$_{rxn}$ | $\checkmark$ | 0.07 | 0.17 | 0.14 | 0.04 | 0.14 |
| *Rsmiles* | | | | | | |
| Rsmiles-unguided | $\times$ | 0.01 | 0.09 | 0.23 | 0.10 | 0.08 |
| Rsmiles-F$_{rxn}$ | $\times$ | 0.09 | 0.19 | **0.52** | 0.05 | 0.19 |
| Rsmiles-F$_{sim}$ | $\times$ | 0.01 | 0.03 | 0.05 | 0.01 | 0.13 |
| Rsmiles-G$_{rxn}$ | $\times$ | 0.11 | **0.21** | 0.47 | 0.10 | 0.19 |
| Rsmiles-G$_{sim}$ | $\times$ | 0.07 | 0.15 | 0.23 | 0.11 | 0.29 |
| *Rsmiles-T* | | | | | | |
| Rsmiles-T$_{rxn}$ | $\times$ | 0.09 | 0.18 | 0.31 | **0.12** | 0.17 |
| Rsmiles-T$_{sim}$ | $\times$ | 0.06 | 0.14 | 0.30 | 0.09 | 0.17 |
| Rsmiles-TG$_{rxn}$ | $\times$ | **0.12** | **0.21** | 0.49 | **0.12** | 0.20 |
| Rsmiles-TG$_{sim}$ | $\times$ | 0.02 | 0.08 | 0.26 | 0.06 | **0.34** |

generation (table 3), while maintaining competitive exact route recovery (12% vs 11% for search guidance) (table 3). These results demonstrate that guiding at the single-step level translates directly to higher-quality multi-step synthesis plans that better satisfy chemist-specified constraints while maintaining or improving solve rates and route diversity.

## 5 RELATED WORK

**Steerable synthesis planning** Steerable synthesis planning has evolved from template-based methods (Segler & Waller, 2017) that encode reaction preferences through rule modifications to neural approaches that operate primarily through search-level interventions. Recent work has focused on reranking (Lin et al., 2022), multi-objective search optimization (Yu et al., 2024; Armstrong et al., 2024), and reinforcement learning approaches (Guo et al., 2024). While systems like AiZynthFinder (Genheden et al., 2020) have demonstrated incorporation of specific constraints, these approaches typically require model retraining or operate through search-space filtering, fundamentally limiting exploration to chemical spaces already accessible by the underlying single-step models. More recently, methods like the LLM-guided search paradigms (Bran et al., 2025) leverage large language models to evaluate and re-rank candidate routes generated by traditional search algorithms based on natural language queries. While powerful for strategic understanding and evaluation, these approaches typically rely on frozen single-step retrosynthesis models and operate by selecting from the already generated candidates, thus fundamentally limiting exploration to chemical spaces accessible by the underlying single-step model.

**Classifier-guidance in generative and autoregressive models** Classifier guidance techniques have proven effective for controllable generation across various domains, from diffusion models (Dhariwal & Nichol, 2021a; Ho & Salimans, 2022) to autoregressive text generation. Methods like PPLM (Dathathri et al., 2020) use gradient-based steering, while FUDGE (Yang & Klein, 2021) applies Bayesian classifier guidance through token-level reranking based on future attribute prediction. Applications to chemical synthesis planning remain largely unexplored. Critically, theoretical understanding of classifier guidance in autoregressive settings remains incomplete, particularly regarding space exploration guarantees in structured generation tasks like chemistry, which we address in this work.

## 6 Discussion and Limitations

Our work demonstrates that direct model guidance enables exploration of chemical spaces previously inaccessible to standard retrosynthesis approaches. By intervening at the token level during sequence generation, we achieve systematic bias toward desired properties while maintaining the flexibility to adapt guidance strength based on synthesis requirements. The theoretical framework we establish provides formal guarantees that guided beam search can access arbitrarily low-probability regions of the chemical space when they satisfy target properties.

Our method faces several important limitations. A fundamental tension exists between maximizing property satisfaction and maintaining chemical feasibility. While our framework allows tunable control through the guidance strength parameter $\lambda$, which we exploit through the parameter selection procedure detailed in section E, determining optimal trade-offs remains challenging and may require domain expertise or adaptive strategies that adjust guidance in real-time based on generation quality. Our approach is inherently limited by the performance of the property predictors—inaccurate classifiers will guide generation toward incorrect regions of chemical space. Additionally, we inherit known challenges in reaction prediction including data quality issues and difficulties in evaluation metrics. The computational overhead of classifier evaluation at each generation step, while manageable, represents a practical consideration for resource-constrained deployments.

Our token-level guidance approach is complementary to recent search-level methods, including LLM-based strategies that excel at interpreting natural language queries and re-ranking complete routes (Bran et al., 2025). While search-level guidance operates on pre-generated candidates, our method produces candidates inaccessible to unguided generators. This suggests a promising integration: LLMs could define precise constraints from natural language that our classifier guidance then enforces during generation, provided data relevant to the constraints is available to train property predictors. By focusing on established seq2seq models, our approach maintains chemical validity while enabling deep steerability, complementing the strategic reasoning of LLM-based systems.

## 7 Conclusion

In this work, we developed a theoretically grounded framework for token-level guidance in retrosynthesis that enables exploration of previously inaccessible chemical spaces during generation. We also demonstrated that direct model guidance produces synthesis routes with significantly higher property satisfaction rates while maintaining competitive solve rates compared to search-level approaches. The broader implications extend beyond synthesis planning to steerable generation in structured domains. As chemical databases continue expanding and computational power increases, we anticipate that hybrid approaches combining the theoretical guarantees of template methods with the flexibility of guided neural generation will become increasingly important. Future directions include multi-objective guidance frameworks, adaptive strategies that adjust guidance strength based on generation quality, and integration with learned cost models for economically-aware route optimization. The principles established here provide a foundation for developing more sophisticated steerable synthesis planning systems that can adapt to the evolving needs of synthetic chemistry.

## 8 Reproducibility Statement

We provide a rigorous proof for our theoretical result on chemical space exploration, and include extensive details about our experiments. We will make the code public under MIT License upon acceptance.

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

# APPENDICES

## A THEORETICAL ANALYSES AND PROOFS

### A.1 REACHING NEW REGIONS OF THE CHEMICAL SPACE

**Theorem 2. (Reaching New Regions of the Chemical Space)** *Let $\beta_k^{(l)}$ be the score of the $k$-th beam (i.e. the beam with the lowest accepted score) at generation step $l$, and $R_\phi = \{r \in \mathcal{V}^L \mid \phi_\alpha(r, p) = 1, p_\theta(r \mid p) < \beta_k^{(l)}\}$ be the set of all property-satisfying sequences with base model probability less than $\beta_k^{(l)}$, and assume $R_\phi \neq \emptyset$. Define $r^* = \arg\max_{r \in R_\phi} p_\theta(r \mid p)$ as the property-satisfying sequence with the highest base model probability. Assume the classifier $p_\alpha$ satisfies for all $l \in [1, L]$:*

$$p_\alpha(\phi_\alpha = 1 \mid r_{\leq l}^*, p) \geq c_1 \tag{8}$$

$$p_\alpha(\phi_\alpha = 1 \mid r_{\leq l}, p) \leq c_2, \text{ when } \phi_\alpha(r, p) = 0 \tag{9}$$

*where $0 < c_2 < c_1 \leq 1$. Then:*

1. *Unguided beam search ($\lambda = 0$) excludes $r^*$ from the final output,*

2. *There exists $\lambda^* > 0$ such that guided beam search with guidance scale $\lambda \geq \lambda^*$ includes $r^*$ in the final output.*

*Proof.* We define cumulative scores at each generation step $l$ as:

$$F(r_{\leq l}) = \sum_{t=1}^{l} \log p_\theta(r_t \mid r_{<t}, p) = \log p_\theta(r_{\leq l} \mid p) \tag{10}$$

For guided generation, the cumulative guided score is:

$$F_{\text{guided}}(r_{\leq l}) = \sum_{t=1}^{l} \left[ \log p_\theta(r_t \mid r_{<t}, p) + \lambda \log p_\alpha(\phi_\alpha = 1 \mid r_{\leq t}, p) \right] \tag{11}$$

Proving statement 1 is straightforward. By definition, $r_{\leq L}^* \in R_\phi$ means $\log p_\theta(r_{\leq L}^* \mid p) < \beta_k^{(l)}$. Since the unguided cumulative score is $F(r_{\leq l}^*) = \log p_\theta(r_{\leq l}^* \mid p) < \beta_k^{(l)}$, sequence $r^*$ is excluded from the beam at step $l$, and therefore from the final output.

To prove statement 2, we need to show that there exists $\lambda^* > 0$ such that at every generation step $l \in [1, L]$, the cumulative guided score of $r^*$ exceeds the cumulative score of the most probable non-property-satisfying sequence in the beam:

$$F_{\text{guided}}(r_{\leq l}^*) \geq F_{\text{guided}}(r_{\leq l}^{\text{hard}}) \quad \forall l \in [1, L], \tag{12}$$

with $r_{\leq l}^{\text{hard}}$ denoting the sequence in the beam with the highest cumulative score at step $l$. Expanding both sides:

$$\log p_\theta(r_{\leq l}^* \mid p) + \lambda \sum_{t=1}^{l} \log p_\alpha(\phi_\alpha = 1 \mid r_{\leq t}^*, p) \geq \log p_\theta(r_{\leq l}^{\text{hard}} \mid p) + \lambda \sum_{t=1}^{l} \log p_\alpha(\phi_\alpha = 1 \mid r_{\leq t}^{\text{hard}}, p)$$

Using the bounds from eq. (8):

$$\sum_{t=1}^{l} \log p_\alpha(\phi_\alpha = 1 \mid r_{\leq t}^*, p) \geq l \log c_1 \tag{13}$$

$$\sum_{t=1}^{l} \log p_\alpha(\phi_\alpha = 1 \mid r_{\leq t}^{\text{hard}}, p) \leq l \log c_2 \tag{14}$$

Substituting:

$$\log p_\theta(r_{\leq l}^* \mid p) + \lambda l \log c_1 \geq \log p_\theta(r_{\leq l}^{\text{hard}} \mid p) + \lambda l \log c_2 \tag{15}$$

Solving for $\lambda$:

$$\lambda \geq \frac{\log p_\theta(r^{\text{hard}}_{\leq l} \mid p) - \log p_\theta(r^*_{\leq l} \mid p)}{l(\log c_1 - \log c_2)} \tag{16}$$

Since this must hold for all $l \in [1, L]$, we need $\lambda$ to satisfy the constraint at the most demanding step. We define:

$$\lambda^* = \max_{l \in [1,L]} \frac{\log p_\theta(r^{\text{hard}}_{\leq l} \mid p) - \log p_\theta(r^*_{\leq l} \mid p)}{l \cdot \log(c_1/c_2)} \tag{17}$$

This ensures that $r^*$ will have sufficient cumulative score to remain in the beam at every generation step. Therefore, with $\lambda \geq \lambda^*$, sequence $r^*$ is guaranteed to be included in the final output. $\square$

## A.2 DELAYED GUIDANCE

**Corollary A.1** (Delayed Guidance)**.** *Under the conditions of theorem 2, suppose the classifier satisfies the error bounds in eq. (8) only for partial sequences with $l \geq l_{\min}$. If $r^*_{\leq l_{\min}}$ remains in the beam during unguided generation (i.e., $\log p_\theta(r^*_{\leq l_{\min}}|p) \geq \beta_k^{(l_{\min})}$), then guided beam search with guidance applied at $l > l_{\min}$ includes $r^*$ in its final output for sufficiently large $\lambda$.*

*Proof.* Since $r^*_{\leq l_{\min}}$ survives unguided beam search up to step $l_{\min}$, we apply Theorem 2's argument for steps $l \in [l_{\min}, L]$. For any step $l \geq l_{\min}$, we require:

$$\log p_\theta(r^*_{\leq l}|p) + \lambda \sum_{t=l_{\min}}^{l} \log p_\alpha(\phi_\alpha = 1|r^*_{\leq t}, p) \geq \log p_\theta(r^{\text{hard}}_{\leq l}|p) + \lambda \sum_{t=l_{\min}}^{l} \log p_\alpha(\phi_\alpha = 1|r^{\text{hard}}_{\leq t}, p) \tag{18}$$

Using bounds and solving for $\lambda$ yields:

$$\lambda \geq \frac{\log p_\theta(r^{\text{hard}}_{\leq l}|p) - \log p_\theta(r^*_{\leq l}|p)}{(l - l_{\min}) \cdot \log(c_1/c_2)} \tag{19}$$

Taking the maximum over $l \in [l_{\min}, L]$ gives the required guidance strength. The remainder follows identically to Theorem 2. $\square$

## B DATASETS AND PROCESSING

We present here the datasets used in this work. For the new datasets introduced, we explain the generation procedure, in addition to the purpose of the dataset and some key statistics.

**USPTO-50k**    First introduced by Schneider et al. (2016), this dataset comprises 50,000 unique reactions randomly selected from US patent documents spanning 1976–2015, originally extracted via text mining by Lowe (2012). The dataset is split into 40,000 training, 5,000 validation, and 5,000 test reactions. Two atom-to-atom mapping approaches were applied: NameRxn (version 2.1.84), which assigns reaction types and mappings using expert-defined SMIRKS patterns, and the Indigo toolkit, configured to ignore charges and valency while allowing bond order changes. The NameRxn mappings were subsequently manually verified by experts for correctness.

**USPTO-190 or USPTO-hard (test set)**    The dataset was compiled by Chen et al. (2020) from synthesis routes constructed from USPTO-Full by identifying the shortest possible routes from each product molecule to a set of commercially available starting materials (23M molecules from eMolecules as of November 2019). Routes were extracted using reactions from either the training set alone, or combined training/validation, or training/test sets of USPTO-Full. The dataset was further filtered to retain only routes where all constituent reactions have solutions within the top-50 predictions of the template-based model NeuralSym (Segler & Waller, 2017), as trained by Chen et al. (2020). Note that we use the NeuralSym implementation and checkpoints provided by Yu et al. (2024), which achieves 65% top-50 coverage matching our results in table 1. The training subset contains 299,202 routes, while validation and test sets contain 65,274 and 189 routes respectively.

**USPTO-190 (training set)**    Here we explain the kind of processing we apply to the USPTO-190 training set to generate the data used to train our RSMILES on the same dataset as Segler & Waller (2017) in (Yu et al., 2024). The dataset initially approx. 4M reactions (extracted from 299,202 routes), with a vocabulary size of 237. In an effort to reduce the vocabulary size needed to train the model, and later apply guidance, we discard all reactions with rare tokens (i.e. tokens present in less than 500 reactions of the dataset). This leaves us with 2M reactions, with a vocabulary size of 53. We further sample 10% of this dataset for our training set, and 0.5% for our validation set, leaving us with 200k reactions for training and 10k reactions for validation.

**Pistachio-Reachable**    Shared by Yu et al. (2024), this dataset contains 150 routes from the proprietary Pistachio dataset, and was sampled out of 1,004,092 test routes extracted following the procedure introduced by Chen et al. (2020) and described earlier. The final samples of routes meets additional criteria set by the authors, including no routes with overlapping reactions, no two targets share a tanimoto similarity greater than 0.7, and the routes belong to certain lengths with pre-defined proportionality.

**Pistachio-Hard**    Also introduced by Yu et al. (2024), this dataset contains 100 routes from the proprietary Pistachio dataset. The difference between this dataset and Pistachio-Reachable is that the routes in this dataset are more difficult to synthesize, by setting a maximum in-distribution rate of 50%.

**Pistachio-Reachable**    Introduced by Yu et al. (2024), this dataset contains 150 synthesis routes extracted from the proprietary Pistachio database (**?**). Routes were sampled from 1,004,092 candidates generated by applying the extraction procedure of Chen et al. (2020) to reactions within Pistachio patents. The final 150 routes satisfy the following criteria: (1) no reactions appear in the USPTO training data, (2) no reactions are shared between routes in the test set, (3) all reactions are recoverable within the top-50 predictions of the NeuralSym retrosynthesis model, (4) no two target molecules have Tanimoto similarity exceeding 0.7, and (5) route lengths follow a specified distribution to ensure representation across difficulty levels. This dataset represents an in-distribution benchmark where 100% of reactions are reproducible by the single-step model used in our work.

**Pistachio-Hard**    Also introduced by Yu et al. (2024), this dataset contains 100 synthesis routes from the Pistachio database following a similar extraction procedure to Pistachio-Reachable. The key distinction is that criterion (3) is relaxed to require only 50% or more of reactions per route to be recoverable within the top-50 predictions of NeuralSym, creating a more challenging out-of-distribution benchmark. The average in-distribution rate across the dataset is 59.9%, compared to 100% for Pistachio-Reachable and 65.6% for USPTO-190. Routes in Pistachio-Hard also exhibit higher reaction uniqueness (95.2% unique reactions vs. 86.1% in Pistachio-Reachable) and greater average depth (7.2 vs. 5.4 steps), reflecting increased complexity.

**USPTO-190-Steps (new)**    We introduce this dataset to serve two purposes: 1) provide a set of realistic reactions for evaluating single-step models, and 2) assess the theoretical upper bound on the recovery of ground truth routes. To this end, we extract the individual reactions from the USPTO-190 routes, and we evaluate the single-step model on them. The dataset contains 640 deduplicated reactions, for which we assign reaction types and atom-mappings using rxn-insight (Dobbelaere et al., 2024). We also compute the tanimoto similarity with the starting material from the route that is most similar to the target, and save said starting material for each reaction.

**Mathematical expressions (toy experiment) (new)**    We generate a synthetic dataset of mathematical expressions with different levels of simplification to validate our guidance framework in a controlled setting. Each training example consists of a complex expression (source, e.g. "(3 + 5) - (2 + 1)") and its simplified form (target, e.g. "5 + 2" or "5 + ( 3 - 1 )"). Note that we intentionally restrain the vocabulary to digits 1 to 3, the operators to +/-, and the level of simplification to 'left-side solved' or 'both-solved' (i.e. not considering the simplification of a single number), to make the guidance effect readily observable. We generate 100k pairs from each complex expression, where each source is paired with two targets, one of which is the 'left-side solved' form, and the other is the 'both-solved' form. The dataset is split into train and test sets using a 80-20 split. The training split is further processed to produce a training set for the classifier, containing partially completed

sequences for each source-target pairs, following the same procedure explained later for chemical property predictors.

**Property data for chemical predictors (new)** We construct training datasets for both reaction type and Tanimoto similarity predictors by generating partially completed precursor sequences at varying completion stages. For each example, we extract the precursor SMILES and generate partial sequences ranging from a minimum completion ratio (default 0.8) to full completion. Each partial sequence is concatenated with its corresponding conditioning context: the product SMILES for reaction type classification, or the starting material SMILES (with separator token) for Tanimoto similarity prediction. Complete sequences are augmented using R-SMILES-style canonicalization with different root atoms (max 5 augmentations). For uspto-190 trained predictors, we use the routes retained from the filtering process described for USPTO-190 (training set), whereas we use the USPTO-50k training set to extract the data for the uspto-50k trained predictors.

| partial sequence [sep] product | reaction type | Ratio |
|---|---|---|
| O=C(c1ccccn1)[…].CC(=O)c>>CC(=O)[…](C(F)(F)F)cccc12 | 2 | 0.80 |
| O=C(c1ccccn1) […].CC(=O)c1>>CC(=O)[…](C(F)(F)F)cccc12 | 2 | 0.81 |
| … | 2 | … |
| O=C(c1ccccn1) […].CC(=O)c1ccccc1B(O)O>>CC(=O)[…](C(F)(F)F)cccc12 | 2 | 1.0 |
| … | 2 | 1.0 |
| c1(C(=O)c2cn[…].OB(O)c1ccccc1C(C)=O>>c12ncc(C([…]c1cccc2C(F)(F)F | 2 | 1.0 |

Augmentation = 5 (for the last three product rows)

| partial sequence [sep] starting material | Tanimoto | Ratio |
|---|---|---|
| O=C(Nc1ccc[…]c(F)c1<unk>Nc1ccc[…]c(F)c1 | 0.8 | 0.80 |
| O=C(Nc1ccc[…])c(F)c1)C<unk>Nc1ccc[…]c(F)c1 | 0.8 | 0.81 |
| … | 0.8 | … |
| c1cc(NC(=O[…](F)c1Oc1ccnc2[nH]ccc12<unk>Nc1ccc[…]c(F)c1 | 0.8 | 1.0 |
| … | 0.8 | 1.0 |
| [nH]1ccc2c([…]NO)C(F)(F)F)cc3F)ccnc12<unk>Nc1ccc[…]c(F)c1 | 0.8 | 1.0 |

Augmentation = 5 (for the last three starting material rows)

Figure 4: The data used to train the reaction type and tanimoto predictors. In both cases, we concatenate the partial complete sequence of precursors with the corresponding conditional string (product or starting material). We fix the minimum completion ratio to 0.8, and augment the complete sequences with 5 root-aligned canonicalizations.

# C PROPERTIES

We discuss here the two properties we choose to guide towards. The discussion includes the definition of the property, its possible values, and why we think it is an interesting application of our method. We also include other ideas for properties that could be of interest for future work, and potential hurdles in their use.

**Reaction type** Reaction types represent mechanistic classifications of chemical transformations based on the bond-forming and bond-breaking patterns, independent of specific reagents or conditions used. We use the reaction classes defined by Carey et al. (2006). This classification system emerged from analyzing 1039 reactions across pharmaceutical R&D departments at GlaxoSmithKline, AstraZeneca, and Pfizer, making it particularly well-suited for drug synthesis applications. We provide a summary of the reaction classes, as well as their distribution in our test sets, in table 4. In practice, we use rxn-insight (Dobbelaere et al., 2024) to assign reaction classes to the reactions in the USPTO-190 dataset, as well as to obtain the reaction type during evaluation and parameter selection. Using this tool has it limitations, however, since it has a reported accuracy of $95.5\%$ in the naming task and $91.1\%$ on USPTO-50k, we believe these limitations are minor.

Table 4: Summary of Carey reaction type classification system showing mechanistic categories and their distribution in pharmaceutical synthesis datasets. Heteroatom alkylation/arylation and C-C coupling dominate (24-33% and 15-16% respectively), while oxidations and functional group additions are rare (1-3%), reflecting the construction patterns typical in drug synthesis routes.

| Reaction type | Description | Index | USPTO-50k | USPTO-190 |
|---|---|---|---|---|
| Heteroatom Alkylation & Arylation | Forms C-N, C-O, C-S bonds via alkylation or arylation | 0 | 0.33 | 0.24 |
| Acylation | Introduces acyl groups (C=O) to heteroatoms | 1 | 0.19 | 0.12 |
| C-C Coupling | Forms carbon-carbon bonds | 2 | 0.16 | 0.15 |
| Aromatic Heterocycle Formation | Constructs aromatic rings containing heteroatoms | 3 | 0.06 | 0.05 |
| Protection | Installs protecting groups on functional groups | 4 | 0.03 | 0.05 |
| Deprotection | Removes protecting groups | 5 | 0.03 | 0.08 |
| Reduction | Decreases oxidation state (e.g., carbonyl to alcohol) | 6 | 0.07 | 0.15 |
| Oxidation | Increases oxidation state (e.g., alcohol to carbonyl) | 7 | 0.01 | 0.03 |
| Functional Group Addition | Adds new functional groups to molecules | 8 | 0.01 | 0.01 |
| Functional Group Interconversion | Converts one functional group to another | 9 | 0.05 | 0.06 |
| Miscellaneous | Other transformations not fitting above categories | 10 | 0.04 | 0.07 |

**Tanimoto similarity**  For starting material-constrained synthesis, we guide the model to generate precursors structurally similar to a specified target starting material. We measure structural similarity using the Tanimoto coefficient (also known as the Jaccard index), a widely adopted metric in cheminformatics for comparing molecular fingerprints (Bajusz et al., 2015). Given two molecules represented as binary Morgan fingerprints $A$ and $B$, the Tanimoto similarity is:

$$T(A, B) = \frac{|A \cap B|}{|A \cup B|} = \frac{|A \cap B|}{|A| + |B| - |A \cap B|} \tag{20}$$

where $|A \cap B|$ counts shared structural features (bits set to 1 in both fingerprints) and $|A \cup B|$ counts the union of features. The coefficient ranges from 0 (no shared substructures) to 1 (identical molecules). We use Morgan fingerprints with radius 2 and 2048 bits to represent molecular structures. In practice, we use `landrum2019rdkit` to compute the Tanimoto similarity between the starting material and the generated precursors.

**Other properties**  While we focus on reaction type and starting material constraints in this work, our guidance framework naturally extends to other reaction properties relevant to synthesis planning. We briefly discuss potential properties of interest and considerations for their implementation:

- **Reaction yield.** Predicted or historical yield data could guide generation toward high-yielding transformations, though yield is highly dependent on specific reaction conditions not captured in SMILES representations. Training data availability, the context-dependent nature of yields, and the lack of a neutral evaluation method, present significant challenges. Previous work (Schwaller et al., 2021) notes the challenge in predicting reaction yields for patent reaction data in particular.

- **Toxicity.** Guiding away from toxic intermediates or reagents could improve process safety and environmental impact. However, toxicity prediction models for reactive intermediates remain limited, and many toxic species are unavoidable in certain synthetic transformations.

- **Synthesizability.** Metrics like synthetic accessibility scores (SA score) (Ertl & Schuffenhauer, 2009) could bias generation toward more readily synthesizable intermediates. This property could help avoid complex intermediates that would themselves require challenging synthesis.

Each of these properties would require training appropriate classifiers or regressors, with the key challenge being availability of reliable training data that correlates property labels with reaction SMILES representations.

## D EXPERIMENTAL SETUP

We provide more details on the experimental setup for the single-step and multi-step experiments.

**Round-trip accuracy**    This is computed using R-smiles' forward model, which was trained on the MIT dataset (Zhong et al., 2022). We choose this model because it has one of the best forward models currently available. During evaluation, we run each generated precursor through the forward model, generate 100 samples per product, and consider the precursor successful if the true products appears in the top-10 predictions of the forward model (essentially using top-10 round trip accuracy).

**Reaction name identification**    This is computed using rxn-insight (Dobbelaere et al., 2024). We use this tool as a complement of the round trip accuracy (i.e. if a reaction is named correctly, it is more likely to be feasible). We report the average number of precursors which have a correct reaction name for each product.

## E PARAMETER SELECTION PROCEDURE

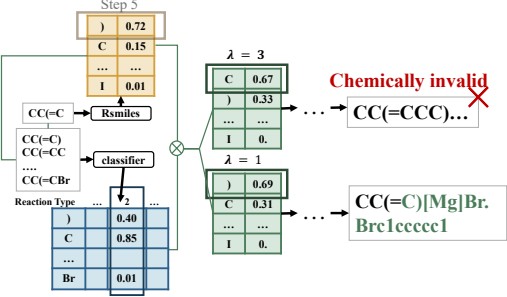

Figure 5: Tuning the guidance scale (and minimum length) is crucial and product-dependent. Excessive guidance ($\lambda = 1$) leads to chemically invalid precursors, while optimal guidance ($\lambda = 0.3$) successfully generates the desired C-C coupling reactants.

**Parameter selection procedure**    The interplay between generator and property predictor logits makes parameter selection product-dependent and critical for performance. As illustrated in Figure 6, excessive guidance can paradoxically degrade output quality: when the generator's top-k tokens already satisfy the desired property, an overly confident classifier may inappropriately rerank toward chemically implausible alternatives that spuriously correlate with high property scores in the partial sequence. Conversely, insufficient guidance fails to redirect the model when needed.

To address this challenge while maintaining fair comparison with baseline methods, we developed a *lookahead parameter selection* procedure (Algorithm 2) that dynamically identifies the optimal guidance strength for each product within a fixed computational budget. The key insight is to allocate a small fraction of the sampling budget to exploration across candidate parameters, then exploit the best-performing parameter for remaining samples.

**Procedure details**    For each product, we evaluate $m$ candidate guidance scales (e.g., $\Lambda = \{0.5, 1.0, 1.5\}$ for reaction type, $\{5, 10, 20\}$ for Tanimoto similarity) by generating $n_{explore} = 10$

---

**Algorithm 2** Lookahead Parameter Selection for Guided Generation

---

**Require:** Product molecule $p$, total sample budget $N$, candidate parameters $\Lambda = \{\lambda_1, \ldots, \lambda_m\}$, exploration samples per parameter $n_{\text{explore}}$
**Ensure:** Set of $N$ generated precursors with optimal guidance
1: $n_{\text{exploit}} \leftarrow N - m \cdot n_{\text{explore}}$ {Reserve samples for exploitation}
2: $\mathcal{P} \leftarrow \emptyset$ {Initialize precursor set}
3: **for** $\lambda \in \Lambda$ **do**
4:    Generate $n_{\text{explore}}$ precursors: $\{r_1^\lambda, \ldots, r_{n_{\text{explore}}}^\lambda\} \sim p_\theta^\lambda(\cdot|p, \phi_\alpha = 1)$
5:    Compute property scores: $s_i^\lambda \leftarrow \phi_\alpha(r_i^\lambda, p)$ for $i = 1, \ldots, n_{\text{explore}}$
6:    $\bar{s}^\lambda \leftarrow \frac{1}{n_{\text{explore}}} \sum_{i=1}^{n_{\text{explore}}} s_i^\lambda$ {Mean property satisfaction}
7:    $\mathcal{P} \leftarrow \mathcal{P} \cup \{r_1^\lambda, \ldots, r_{n_{\text{explore}}}^\lambda\}$
8: **end for**
9: $\lambda^* \leftarrow \arg\max_{\lambda \in \Lambda} \bar{s}^\lambda$ {Select best parameter}
10: Generate $n_{\text{exploit}}$ additional precursors with $\lambda^*$: $\{r_1^*, \ldots, r_{n_{\text{exploit}}}^*\} \sim p_\theta^{\lambda^*}(\cdot|p, \phi_\alpha = 1)$ {Exploitation phase}
11: $\mathcal{P} \leftarrow \mathcal{P} \cup \{r_1^*, \ldots, r_{n_{\text{exploit}}}^*\}$
12:
13: **return** $\mathcal{P}$

---

samples per parameter. We then select $\lambda^* = \arg\max_\lambda \bar{s}^\lambda$ where $\bar{s}^\lambda$ is the mean property satisfaction across exploration samples, and generate the remaining $N - m \cdot n_{\text{explore}}$ samples using $\lambda^*$. This allocates only $m \cdot n_{\text{explore}}/N$ of the budget to exploration (e.g., 30% for $m = 3$, $n_{\text{explore}} = 10$, $N = 100$), ensuring fair comparison with baselines under identical computational constraints.

**Design rationale**   This approach offers several advantages over alternatives:

- *Product-adaptive*: Unlike fixed global parameters tuned on validation sets, lookahead adapts to the specific generator-classifier dynamics for each target molecule, accounting for variation in how strongly different products require guidance.

- *Budget-neutral*: By incorporating exploration cost into the total sample count, we maintain fairness when comparing against methods that use fixed parameters or no guidance.

- *Generalizable*: The procedure requires no prior knowledge of product properties or similarity to training data, enabling zero-shot application to novel targets.

- *Interpretable exploration-exploitation trade-off*: The ratio $n_{\text{explore}}/N$ explicitly controls the balance between parameter search and sample quality, which can be adjusted based on problem requirements.

Alternative approaches could include validation-set-based parameter selection (requiring held-out data with similar characteristics), learned parameter predictors conditioned on product features (requiring additional model training), or adaptive methods that adjust $\lambda$ during generation based on intermediate scores. However, we find that lookahead strikes an optimal balance between simplicity, generalizability, and effectiveness for the synthesis planning domain where products often differ substantially in their chemical properties and how they respond to guidance.

# F  TRAINING PROPERTY PREDICTORS

We train the property predictors by generating partially completed sequences for each task, and training a regression or classification model (depending on the property) to predict the target values from the given partial sequence.

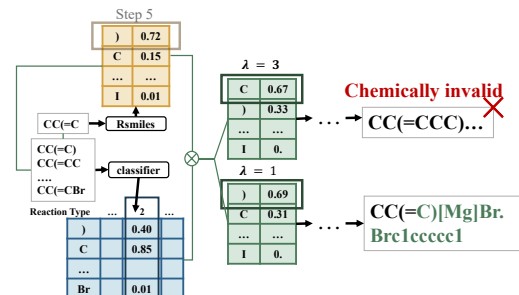

Figure 6: Tuning the guidance scale (and minimum length) is crucial and product-dependent. Excessive guidance ($\lambda = 1$) leads to chemically invalid precursors, while optimal guidance ($\lambda = $

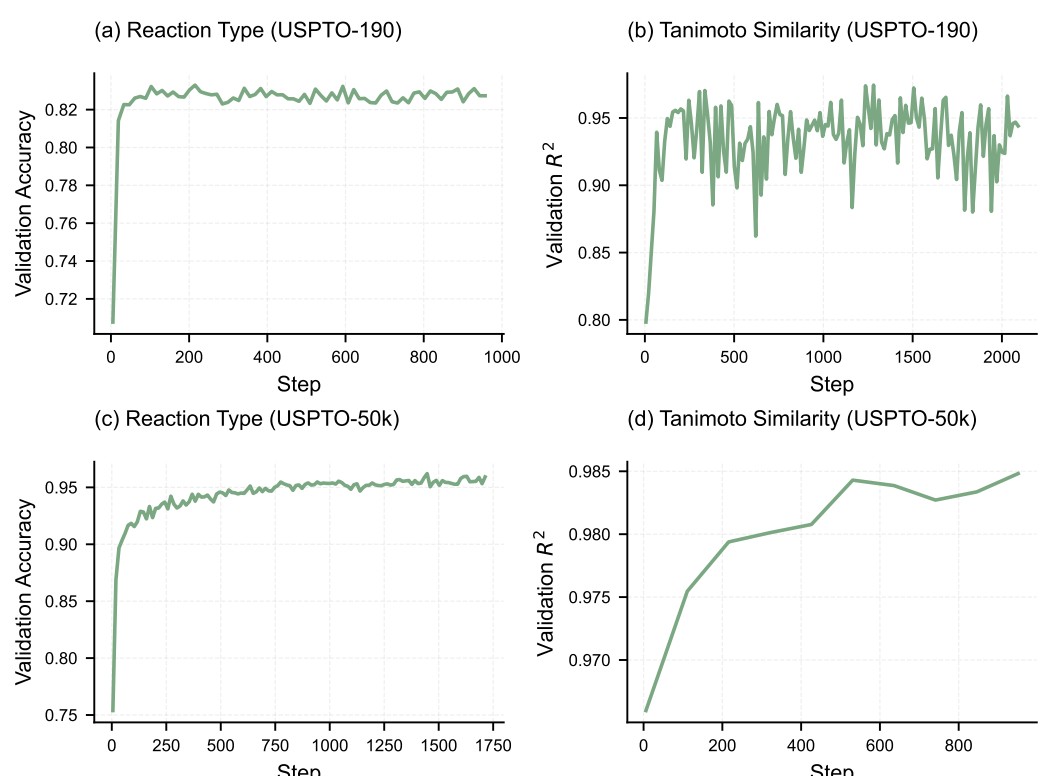

Figure 7: Validation curves for the reaction type and tanimoto similarity classifiers on USPTO-50k and USPTO-190.

**Architecture** All predictor neural networks are a vanilla transformer encoder with 4 layers, 4 attention heads, 128 hidden dimensions, and 0.1 dropout.

**Training details** For optimization, we employ Adam optimizer with a learning rate of 1e-3 and weight decay of 1e-5. We train all models for 2000 epochs, which takes 4 hours on a single A100 GPU. We use the training dataset described in section B for each property predictor. The performance of the models on a validation set is reported in fig. 7.

**Checkpoints** We use the checkpoints at epochs 100 and 210 for tanimoto and reaction type predictors respectively.

## G ILLUSTRATING THE EFFECT OF CLASSIFIER-GUIDANCE IN A TOY EXPERIMENT

We describe a small-scale experiment showcasing guided generation in an autoregressive sequence-to-sequence model. The task is simplifying mathematical expressions, with guidance based on the length of the simplified result. We constrain the vocabulary to digits 1 to 3 and operators +/- to make the guidance effect readily observable. Given an expression like ( 3 + 2 ) + ( 3 - 1 ), there are two valid simplifications: partial simplification 5 + ( 3 - 1 ) (length 7 tokens) and full simplification 5 + 2 (length 3 tokens). A well-trained seq2seq model can produce either

output. To control which simplification the model generates, we train a classifier to predict target sequence length from partial token sequences. During inference, we use the classifier's logits to rerank the seq2seq model's beam search candidates, guiding toward the desired length class. This demonstrates how property predictors can guide generation toward specific outcomes.

**Methods** We generate mathematical expressions and their algorithmic simplifications for training data. Tokens are space-separated substrings (numbers, operators, parentheses). For the classifier, we create partial sequences from target outputs with their full length as the class label. For example, from `5 + ( 3 - 1 )` (class 7), we generate prefixes `5 +`, `5 + (`, `5 + ( 3`, `5 + ( 3 -`, etc. We balance classifier data via undersampling since length-7 sequences were overrepresented. The classifier uses a vanilla transformer with classification head; the seq2seq model uses an ONMT transformer decoder. For evaluation, we test on 20k expressions with and without guidance. Without guidance, either simplification counts as correct. With guidance, only the target class counts as correct.

**Additional setup** Training uses 80k samples (both models) with 20k test samples. The classifier has 6 layers, 8 heads, 16 hidden dimensions, 0.1 dropout, trained 2000 epochs (lr=1e-3, weight decay=1e-5, Adam optimizer, 4 hours on A100). The seq2seq model has 16 layers, 256 embedding size, 1024 transformer_ff, trained 6000 epochs (lr=0.5, weight decay=1e-5, 4 hours on A100) until reaching 90% validation accuracy.

**Results** We first report the results without guidance. Our seq2seq model achieves a 84% accuracy, of which 25% are longer sequences. When guiding the model towards the longer simplification, we recover the expected class in 88% of the cases, while we get 73% accuracy for the short simplification, which shows that we can indeed steer the model towards the desired class with high accuracy. We visualize what happens on a low scale by plotting the original scores from the seq2seq model, the predictions of the classifier, and the final combined scores for one particular example in fig. 8. .

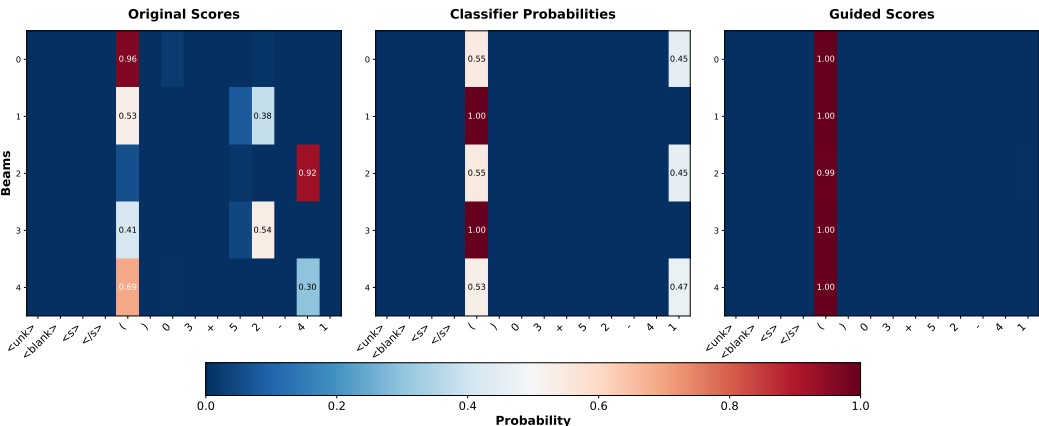

Figure 8: The effect of the classifier logits on the ranking of next token predictions of the seq2seq model. The figure shows the result of translating `( 2 + 3 ) + ( 1 + 3 )` with guidance towards the longer simplification (class 7), with a guidance scale of 0.3 and a beam size of 5. The leftmost column shows the original scores from the seq2seq model, the middle column shows the predictions of the classifier, and the rightmost column shows the final combined scores. Notice how with the classifier logits, all the beams are biased towards the correct next token in this case `(`.

We found that suitable values for the guidance scale are between 0.1 and 1.0, and that the guidance is more effective when the minimum length for guidance is set to 3 (as in no guidance will be applied for the first 3 tokens). To discourage the seq2seq model from producing short sequences, we add an eos penalty of -10 to the logits of the model during beam search. Since the earlier tokens contain the most uncertainty, we train some models without them.

