# OpenReview forum: "Beyond Search: Direct Model Guidance for Steerable Synthesis Planning"
_ICLR.cc/2026/Conference — Submitted to ICLR 2026_

### Official Review · Reviewer_nNwF · 2025-10-31

**Soundness:** 1
**Presentation:** 2
**Contribution:** 2
**Rating:** 2
**Confidence:** 4

**Summary:**

This paper investigates guiding a trained single-step retrosynthesis model with the outputs from a separate property predictor. Authors investigate a toy setup as well as two retrosynthesis setups (one based on guiding with reaction class information, and one based on guiding towards the use of a particular building block molecule).

**Strengths:**

**(S1)**: Retrosynthetic search is a timely and important problem to work on. Current search algorithms are often not very steerable, and thus looking into giving more fine-grained control to the user can be useful.

**Weaknesses:**

**(W1)**: Several parts of the work are quite unclear to me:

- **(W1a)**: Authors talk about "guiding towards reaction type", but it's not always clear _which_ reaction type is the target. In single-step and manual multi-step experiments, are predictions simply guided towards the _ground-truth_ reaction type? How does this generalize to a search-based multi-step setup, where the algorithm may end up using reactions outside of training data for which ground-truth (expected) reaction type is not known? Are all reactions during search guided towards the same reaction type in this case?

- **(W1b)**: Does guiding towards a specific starting material mean forcing the predicted reaction to have the required molecule as _one_ of the reactants, allowing potential further reactants to be generated freely? In Section 4.3, authors state this setup works by "forcing the model to ouput the tokens matching the conditional starting material, by (…) increasing the guidance scale to the point of overwhelming the generator logits". Does that mean simply teacher-forcing the autoregressive generator to start the generation with outputting the required reactant? Also, in experiments, authors seem to assume access to ground-truth route depth, and then force the starting material at the given depth; I'm not sure how can this work for convergent routes where there are several reactions at that depth.

- **(W1c)**: If I understand Equation 1 correctly, it says a guided route would satisfy the requirement of having more reactions pass _at least one_ of the property requirements than a route found without guidance. This seems like a confusing formulation, as it requires including a "baseline" route to compare to, doesn't reward cases when a single reaction satisfies several (or all) property predicates simultaneously (even though in experiments author seem to only use a single predicate?), and only provides a minimum standard rather than an objective to optimize. Later sections in the text suggest the objective authors case about is maximizing the expected number of accepted reactions, in which case I'm not sure what purpose Equation 1 serves.

**(W2)**: On top of the clarity issues above, I don't think the setup used in this work is practical. In essence, the authors show that if some aspect of the ground-truth or expected solution is known (e.g. reaction type used, or one of starting materials), guiding the predictor towards that can increase the likelihood of finding that expected solution; this sounds reasonable. However, it raises two questions: (a) whether this approach is practically useful or needed; and (b) whether it is effective in producing good outcomes. I have doubts about both fronts:

- **(W2a)**: Authors look at two potential applications: guiding towards reaction type, and guiding towards using a particular starting material (building block). For the former, I am not sure how this works in an end-to-end search setup (see **(W1a)**), are all steps in the search being guided towards the same reaction type? While chemists may have some preferences (which could be caused by them having explored a particular set of reaction types exhaustively and thus having built intuition or specialized feasibility models for those kinds of reactions specifically), this would more often be a set of reaction types; also, if they had a strong preference to only use a limited set, they may not even use a retrosynthetic search tool to begin with, which are best suited for broad exploration. Therefore, I'm not sure if this setting would be that popular in practice. For guidance towards a specific starting material, I assume the hypothetical setup is when a molecule that is complex but similar to the target has already been synthetized at a particular lab, and thus chemists are looking for routes that reuse that complex intermediate. In that case, a simple baseline would be to include it in the building block set; this would be much simpler, would not require knowing ground-truth depth (see **(W1b)**), and would trivially generalize to having several such intermediates instead of just one.

- **(W2b)**: As stated by authors, there is a trade-off between following the guidance and quality of the predictions. In fact, deterioration seems quite visible in the quantitative results; e.g. in the single-step USPTO-50K experiment, while guidance allows to recover the ground-truth for products where the unguided model fails, the guided model performance is then worse on other products. This is odd to me: assuming my understanding in **(W1a)** is correct, guidance towards ground-truth reaction type should generally only improve results, unless the guiding property model is confusing the generator. In this case, better results would likely be obtained by simply sampling more outputs from the unguided generator and post-filtering, which would then invalidate the need for guidance in the first place.

**Other (much less important) comments**

**(O1)**: In the toy task, vocabulary is constrained to `{1, 2, 3}`, but then larger digits are allowed to appear during simplification, so I'm not sure why the "full simplification" option is not just `7` instead of `5 + 2`.

**(O2)**: While the paper includes some theoretical analysis (Theorem 1), I do not include it as a strength, as I think the result is not that interesting. It is of course nice to have, but I think it doesn't help against my concerns, especially ones around practicality of the setup **(W2)**.

**Questions:**

See the "Weaknesses" section above for specific questions.

---

> ### Author Response · Authors · 2025-11-26
> **Response to nNwF**
>
> We thank the reviewer for their detailed feedback. Below we address their concerns in turn.
>
> # (W1a) Target Reaction Type (Ground-Truth vs. Desired)
>
> In both our single-step (Section 4.1) and multi-step evaluation (Section 4.2) experiments, the guidance for reaction type is indeed set to the ground-truth reaction type for evaluation purposes (lines 339-340). This allows us to quantitatively measure our method's ability to recover known routes and generate precursors for a specific, verified reaction type. While our experiments use ground-truth for benchmarking, the method itself is designed to be steerable towards any user-specified reaction type (or other property), without requiring the target to be a known ground truth.
>
> # (W1a) Guidance in Multi-step Search
>
> No, not all reactions during search are guided towards the same reaction type. Our framework explicitly supports guiding individual steps in a multi-step synthesis towards different desired reaction types. This is clearly illustrated in Figure 1 ( c), where Step 1 is guided towards a 'square' type and Step 2 towards a 'triangle' type, demonstrating the flexibility to enforce a "series of desired reaction types" across a synthesis pathway (lines 301-302).
>
> # (W1a) Generalization and Guidance in Multi-step Search
>
> The reviewer's question regarding "novel reactions" is interesting. In our multi-step search, the "target reaction type" for guidance is typically a pre-defined constraint provided by the chemist as a rough sketch of the route. For example, a chemist might specify that "Step 1 should be a C-C coupling, followed by a reduction in one branch and an oxidation in another." In this scenario, we programmatically assign the desired reaction type (e.g., type 2 for C-C coupling, type 6 for reduction, type 7 for oxidation from Table 4) to the appropriate nodes based on search depth and branching factors. If the search explores beyond this pre-defined sketch, guidance for reaction type is not applied for those deeper or more divergent nodes.
>
> # (w1b) starting material guidance setup: teacher forcing
> We thank the reviewer for this important clarification request. We have significantly revised our approach to starting material-constrained synthesis in the updated manuscript.
> - **Clarification on the constraint**: Yes, starting material constraint means the synthesis route must incorporate a specific target starting material (e.g., utilizing waste products or renewable feedstocks), while other reactants can be freely selected. This addresses scenarios where chemists want to synthesize a target molecule specifically from a given precursor.
> - **Major methodological change**: We have removed the teacher-forcing approach mentioned in the previous version, as we agree it does not properly demonstrate classifier guidance capabilities. Instead, we now guide the single-step model toward precursors with high Tanimoto similarity to the desired starting material. The intuition is that by maximizing structural similarity at each retrosynthetic step, the search progressively generates intermediates closer to the target starting material until it is eventually incorporated into the route.
> - **Ground truth depth assumption**: We no longer assume access to ground truth route depth. The Tanimoto similarity guidance operates at each expansion during search without requiring knowledge of where in the route the starting material should appear. The search algorithm naturally determines the appropriate depth based on structural similarity scores.
> - **Convergent routes**: For routes with multiple reactions at the same depth, our approach guides each expansion independently toward higher Tanimoto similarity with the target starting material. The search explores multiple branches, and any branch that successfully incorporates the starting material (Tanimoto = 1.0) satisfies the constraint.
>
> We have updated Section 3.4, Algorithm 1, and the experimental sections to reflect these changes.
>
> # (w1c) Confusing equation 1
> Thank you for this valuable feedback. You are correct that Equation 1 was confusing and did not clearly convey our objective. We have substantially revised Section 3 to address these concerns. Changes made include:
> - **Removed Equation 1 entirely**: We agree it was unnecessarily complex and relied on comparison with a baseline route, which was not how we actually implemented or evaluated our method.
> - **Added Section 2.2**: We now provide a clear mathematical formulation of classifier guidance for autoregressive models (Equations 1-4), establishing the theoretical foundation before describing our specific application.
> - **Rewrote Section 3.1**: The problem formulation now focuses on our actual objective: maximizing the expected property satisfaction for each single-step prediction (Equation 5). This directly addresses your point about optimizing the expected number of accepted reactions rather than comparing to a baseline.

---

> ### Author Response · Authors · 2025-11-26
> **Response - continued**
>
> # (w1c) Guiding towards multiple predicate
> Our current experiments indeed only guide towards a single predicate. We are working on providing an experiment combining both predicates promptly.
>
> # (W2)[...] the authors show that if some aspect of the ground-truth or expected solution is known [...], [...] can increase the likelihood of finding that expected solution
>
> We appreciate the reviewer raising this important point, as it allows us to clarify the broader scope and contributions of our work beyond what W2 suggests.
> - Our claim is fundamentally broader than recovering known ground-truth solutions. While we do demonstrate improved ground-truth recovery empirically, our theoretical and methodological contributions establish that token-level guidance enables:
> - **Steering toward arbitrary practitioner-specified properties:**. Our framework (Section 3.4) accommodates any measurable reaction property with available training data, including reaction types, starting material constraints, and extensible to yield prediction, toxicity minimization, or cost optimization. The practitioner need not know the ground truth - only the desired constraint.
> - **Accessing previously unreachable chemical space:** Theorem 1 (Section 3.3) provides the first formal guarantee that classifier-guided beam search can provably reach regions of the output space that remain inaccessible to unguided generation regardless of beam width. This is a fundamental reachability result, not merely an empirical improvement in recovery rates.
> - **Generating chemically valid precursors absent from base model distributions** Our single-step evaluation (Table 1, Figure 3) demonstrates that guided samples exhibit only 39% Jaccard overlap with unguided samples, and among 124 products where unguided R-SMILES fails completely, guidance recovers 55% via round-trip accuracy. These are genuinely new, chemically valid precursors, not just reranking of existing candidates.
>
> # (w2a) whether this approach is practically useful or needed (reaction type)
> We appreciate the reviewer's questions about practical applicability. We'd like to clarify two important aspects of our approach:
>  - **Multi-step guidance with multiple reaction types**: The reviewer asks whether all steps use the same reaction type. As illustrated in Figure 1 (panel c) and discussed in Section 3.4, our framework supports different reaction types at each step of the synthesis route. A chemist can specify a sequence of preferred reaction types (e.g., C-C coupling for step 1, reduction for step 2), not a single constraint applied uniformly. This mirrors realistic synthesis planning where chemists reason about strategic disconnections at different stages.
>  - **Partial solutions as practical outcomes**: The reviewer questions whether chemists with strong preferences would use retrosynthetic tools at all. This mischaracterizes our contribution. The goal is precisely to enable chemists to explore how their preferences integrate with computational suggestions. Our method produces partial solutions that satisfy constraints to varying degrees, for example, routes that incorporate the desired starting material but require additional steps, or routes that use preferred reaction types for key transformations while exploring alternatives elsewhere. These partial solutions help chemists evaluate the feasibility of their strategic choices and identify promising alternatives, which is valuable even when complete solutions matching all preferences do not exist.

---

> ### Author Response · Authors · 2025-11-26
>
> # (w2a) whether this approach is practically useful or needed (starting material constraint)
> The reviewer raises an important point about practical applicability. We clarify two key aspects:
> - **Use case motivation:** Starting material-constrained synthesis addresses waste valorization and resource optimization, not just reusing complex intermediates. As demonstrated by [1], repurposing underutilized chemical waste into valuable products is a significant practical concern. Chemists may have access to abundant but underutilized compounds and seek diverse routes incorporating them, rather than simply finding any route that uses them.
> - **Why guidance vs. simply adding to building blocks**: The starting material is already included in the building block set in our experiments (as stated in our problem formulation, Section 2.1). The baseline approach the reviewer suggests would indeed find some routes using it, but:
> - **Diversity of routes**: Our method generates multiple diverse routes incorporating the starting material (Table 2 shows 6.2 non-overlapping routes per target with guidance vs. 5.1 for search-level methods), while simply adding it to building blocks provides no mechanism to favor or diversify routes using it.
>  - **Incorporation likelihood**: Without guidance, whether the starting material appears in discovered routes depends entirely on whether it happens to be among the top-ranked single-step predictions. Our results show only 8% of unguided routes incorporate the target starting material (Table 3), versus 34% with our method.
>
> The key contribution is enabling chemists to deliberately explore the space of routes incorporating specific materials, not just hoping they appear incidentally during search.
>
> # (W2b) effectiveness in producing good outcomes:
>
> We thank the reviewer for this important observation. We have substantially revised our experimental setup to address this concern, and the trade-off is now minimal. Key improvements include:
> - **Parameter selection procedure (Algorithm 2, Appendix E)**: We now use a look-ahead algorithm that adaptively selects optimal guidance strength per product. This ensures guidance helps when needed while avoiding over-guidance that degrades chemical validity.
> - **Updated results**: With this approach, Table 1 shows that token-guided R-SMILES on USPTO-190-Steps maintains competitive accuracy (31% exact route recovery vs 28% unguided) while achieving 23% correct reaction class vs 3% unguided. Critically, round-trip accuracy remains high (11% vs 2% unguided), indicating chemically valid outputs.
> - **Regarding the comparison to "sampling more + filtering"**: We are preparing additional experiments to demonstrate that guidance accesses genuinely different chemical space rather than simply finding rare samples. Figure 3b already shows that guided samples have only 39% Jaccard overlap with unguided samples, and Figure 3a demonstrates that among 124 products where unguided R-SMILES fails, guidance recovers 55% via round-trip accuracy, evidence that these precursors are inaccessible to unguided generation regardless of sample count.
>
> # References
>
> [1] "Computer-designed repurposing of chemical wastes into drugs", Wolos et al., 2022

---

### Official Review · Reviewer_ZoTP · 2025-10-31

**Soundness:** 3
**Presentation:** 3
**Contribution:** 2
**Rating:** 2
**Confidence:** 4

**Summary:**

This paper introduces a method that steers retrosynthesis generation by modifying the token-level logits of a pretrained seq2seq model (RSMILES) through a lightweight property classifier.
The approach aims to make synthesis planning steerable—conditioning on reaction type or specific starting materials—without retraining, and includes a theoretical guarantee that guided beam search can reach low-probability yet property-consistent regions of chemical space.
Conceptually, the idea is promising and well-motivated, bridging diffusion-style classifier guidance with chemical synthesis planning.
However, the experimental and evaluation sections remain thin: the baselines are incomplete, the datasets limited, and the performance gains difficult to attribute purely to the proposed mechanism.

**Strengths:**

### Strengths

1. **Realistic and well-motivated application scenario**
   The paper is driven by an actual need in synthesis planning, where chemists often want to control reaction types or enforce the use of specific starting materials rather than simply find any feasible route. This makes the problem formulation genuinely insight-driven and practically relevant, rather than an artificial controllability setup.

2. **Potentially general framework**
   Although the experiments are conducted only on a single retrosynthesis model (RSMILES), the proposed guidance mechanism that modifies token-level logits through a property classifier appears broadly applicable to other autoregressive chemical or molecular generation models. The idea could in principle be extended to different model architectures or even to other domains where conditional generation under structural constraints is desirable.

**Weaknesses:**

### Weaknesses

1. **Incomplete Baselines and Dataset Coverage**
   The paper compares guided RSMILES only against unguided RSMILES and NeuralSym on USPTO-50k / USPTO-190. However, the field already includes many recent and competitive baselines for constrained or steerable synthesis planning, which are not covered here. Furthermore, all experiments are restricted to USPTO-style datasets; evaluation on larger and noisier benchmarks such as Pistachio would be necessary to demonstrate generalization beyond USPTO distributions.

2. **Dependence on Pseudo-Labels and Unfair Evaluation Setting**
   The reaction-type guidance relies on RXN-Insight-generated pseudo-labels for USPTO-190. These labels are potentially noisy and highly imbalanced, yet the paper does not quantify their accuracy or analyze class-wise performance. More importantly, the guided model effectively receives extra oracle information (the reaction class) that baseline models do not, making the comparison unfair. The authors should either allow baselines to use the same reaction-type hints (e.g., via reranking or constrained search) or explicitly frame the task as conditional retrosynthesis. It would also strengthen the study to verify whether guided generations genuinely fall into subclasses or finer-grained reaction families rather than only matching top-level class labels.

3. **Unclear and Possibly Test-Time-Tuned λ Selection** The guidance strength λ is chosen from a small grid (0.5, 1.0, 2.0) “based on average reaction-type accuracy,” but the paper does not specify which data split is used for tuning or whether the same λ applies across datasets. Several experiments are conducted on failure subsets of the test set (e.g., USPTO-50k failed products), suggesting possible test-time hyperparameter tuning, which could inflate reported improvements. Since the method is promoted as “no retraining and easy to apply to new constraints,” the paper should either demonstrate robustness to λ or propose a principled, validation-free way to choose it.

**Questions:**

None

---

> ### Author Response · Authors · 2025-11-26
> **Response to ZoTP**
>
> We thank the reviewer for acknowledging the importance of our research topic and the merits of our proposed method. Below we address their concerns.
>
> # Additional Baselines
>
> We have expanded our comparison to include all retrosynthesis models available in Syntheseus [1], addressing the concern about incomplete baselines. Our evaluation now covers both template-based methods (LocalRetro, RetroKNN, NeuralSym) and template-free approaches (MHNReact, MEGAN, Graph2Edits, Chemformer, R-SMILES). The results demonstrate that token-guided R-SMILES achieves superior performance across multiple metrics: 31% exact route recovery on USPTO-190-Steps (vs. 16% NeuralSym), 92% multi-step solve rate (vs. 47% NeuralSym), and 42% recovery when guiding toward starting materials. Critically, guided samples show only 39% overlap with unguided samples (Figure 3), providing direct evidence of genuine exploration rather than reranking. Among 124 products where unguided R-SMILES fails, token guidance recovers 55% via round-trip accuracy, demonstrating access to previously inaccessible chemical spaces. Full details are provided in Sections 4.1 and 4.2.
>
> # Extended Dataset Evaluation
> We acknowledge the limitation of evaluating exclusively on USPTO-style datasets. To demonstrate generalization beyond USPTO distributions, we are currently conducting additional experiments on USPTO-50k for single-step retrosynthesis and Pistachio for multi-step planning. These results will be be made available promptly.
>
> # Discussing the limitations of rxn-insight's labels
> We acknowledge this important limitation. RXN-INSIGHT reports 91% classification accuracy on a 50k-reaction benchmark ([2], 2024, Table 2), though this is tested on NameRXN-labeled data which may differ from USPTO-190's distribution.
> The 91% accuracy suggests labels are sufficiently reliable for training property predictors, though we agree that quantifying error propagation from noisy labels to downstream performance would strengthen the evaluation. We've added this as a limitation in Section 6.
>
> # Similarly-constrained baselines
>
> We appreciate the reviewer raising this important methodological point. We have now substantially revised our experimental setup to address this concern and ensure fair comparison. The comparison is now between: **Generate-then-filter (baselines)**: Sample broadly, discard non-matching, and **Guided generation (ours)**: Bias sampling toward property-satisfying regions. We have added explicit language in Section 4.1 and 4.2 clarifying this experimental design. Specifically, all baselines generate 100 samples per product, which are then filtered to retain only precursors satisfying the target property (matching reaction type OR Tanimoto similarity ≥ 0.2 to the starting material). This applies equally to all baselines, while for ours we use a simple look-ahead scheme to choose the right parameter (see **Choice of $\lambda$** below for additional information).
>
> # Finer-grained class analysis and per-class results
> We are working towards providing these results promptly.
>
> # Unclear choice of $\lambda$
> We thank the reviewer for a chance to clarify this point. In our updated experimental setup, $\lambda$ is selected independently for each product using a lookahead procedure on the test set itself. Specifically, we generate 10 samples with each of 3 candidate $\lambda$ values ($\lambda$ ∈ {0.5, 1.0, 1.5} for reaction type; $\lambda$ ∈ {5, 10, 20} for Tanimoto similarity), select the $\lambda$ achieving highest average property satisfaction on these 30 exploratory samples, then generate 70 additional samples with the selected $\lambda$. This totals 100 samples per product, matching the computational budget of all baseline methods. Therefore $\lambda$ is product-specific, not dataset-specific. This is necessary because different products have different generator-classifier dynamic, some require stronger guidance while others perform better with moderate values (illustrated in Appendix E).
>
> # Several experiments are conducted on failure subsets of the test set, [...]
> we now report results on the entire USPTO-190 datasets, and relegate the failure set analysis to a comparison in figure 3.a, which shows that, if choosing to apply guidance to the set of products the model failed on only, we can recover the ground truth product for $10\%$ of them, and find at least one round-trip correct precursor for $55\%$.
>
> # References
>
> [1] "Re-evaluating Retrosynthesis Algorithms with Syntheseus", Maziarz et al., 2023.
>
> [2] "Rxn-INSIGHT: fast chemical reaction analysis using bond-electron matrices", Dobbelaere et al., 2024.

---

> ### Comment · Reviewer_ZoTP · 2025-11-28
>
> Thank you for the detailed author response. I appreciate the effort to address several of the concerns I raised, and I am glad to see the additional experiments and clarifications.
>
> I will keep my original score. While I find the core idea interesting, my assessment is based on the completeness of the current submission. Plus updating my score from 2 to 4 would not materially affect the acceptance decision at this stage.
>
> Thank you again for the thoughtful response.

---

### Official Review · Reviewer_tWiM · 2025-11-01

**Soundness:** 2
**Presentation:** 1
**Contribution:** 2
**Rating:** 2
**Confidence:** 3

**Summary:**

This paper presents a steerable synthesis planning framework that, rather than improving only the search over candidates produced by a fixed single-step retrosynthesis model, directly modifies the token-level decoding of an autoregressive seq2seq model via a learned property predictor. By injecting a guidance term into the logits during generation, the method biases single-step predictions toward reactions or precursors that satisfy user-specified constraints. The authors further provide a theoretical argument suggesting that this token-level guidance can make parts of the chemical space reachable that standard beam search would not explore.

**Strengths:**

- Demonstrates that steering at the generation level (i.e., changing what the single-step model emits) can unlock routes that search-level steering alone would miss.
- Uses a simple, model-agnostic guidance formulation that can in principle support multiple constraint types (reaction class, starting material, etc.) without retraining the base model.

**Weaknesses:**

- Novelty currently oversold. Classifier- or property-based token-level guidance for autoregressive decoding is well established in controllable text generation, and “template-free multi-step retrosynthesis with search” is a mature direction. The main contribution here is adapting that guidance paradigm to neural retrosynthesis and formalizing its accessibility claim—valuable, but not as fundamental as the introduction implies.
- Experimental coverage is too narrow for the main claim. Most results are against (i) vanilla rSMILES and (ii) neuralsym + Retro*, often with the authors’ own models. To substantiate the claim of reaching “previously inaccessible” routes, the paper should either (a) add stronger template-free neural baselines under the same search/computation budget, (b) include a controlled comparison with an LLM-steered or tool-augmented planner, or (c) show steering under multiple simultaneous constraints (e.g., reaction class + inventory + maximum depth).
- Related work is incomplete. The paper largely contrasts with search-level/ranking methods (Segler et al., Lin et al., Tango*) but omits recent LLM-driven, chemistry-aware steering approaches that target the exact same “interactive, preference-informed synthesis” use case—e.g. Bran, Andres M., et al., “Chemical reasoning in LLMs unlocks steerable synthesis planning and reaction mechanism elucidation,” arXiv:2503.08537 (2025). This line should be acknowledged and the differences in controllability, data requirements, and integration with planners should be discussed.

**Questions:**

N/A

---

> ### Author Response · Authors · 2025-11-26
> **Response to tWiM**
>
> We thank the reviewer for their valuable feedback. Below we address their concerns and answer their questions:
>
> # Overselling the contribution
>
> We appreciate this feedback, and we revised the introduction to better position our contribution. We agree that classifier guidance for autoregressive models is established in text generation. Our core contributions are:
> - **Domain adaptation**: Applying token-level guidance to retrosynthesis, where molecular validity constraints and reaction feasibility requirements present challenges absent in text generation
> - **Theoretical guarantees**: Providing the first formal proof (Theorem 1) that classifier-guided beam search can provably access chemical spaces unreachable by unguided search, a reachability analysis not present in prior guidance literature [1].
> We have modified the introduction (lines 38-48) to clarify that we are adapting an established technique to a new domain with novel theoretical analysis, rather than presenting the guidance mechanism itself as fundamentally new.
>
> # Experimental coverage is too narrow
> We have substantially expanded our experimental scope beyond the initial submission:
> -**Comprehensive Baseline Comparison (addressing point a)**: We now compare against all retrosynthesis models available in Syntheseus [2] under equivalent computational budgets. This includes:template-free methods (MHNReact, MEGAN, Graph2Edits, Chemformer, R-SMILES) and template-based methods: LocalRetro, RetroKNN, NeuralSym.
> - **Single-Step Results (Table 1) demonstrate access to previously inaccessible routes**:
>     - 31% exact route recovery on USPTO-190-Steps (vs. 28% unguided R-SMILES, 16% best baseline NeuralSym)
>     - **Critical evidence of new chemical space**: Guided samples show only 39% overlap with unguided samples (Figure 3), confirming we explore genuinely different reaction pathways rather than reranking existing proposals
>     - Among 124 products where unguided R-SMILES fails completely, token guidance recovers 55% via round-trip accuracy and 10% via exact match (Figure 3)
> - **Multi-Step Planning (Tables 2-3) compared to controlled synhtesis planner (b):**
>     - Reaction type guidance: 92% solve rate, 49% type adherence (vs. 81%/23% unguided, 47%/— NeuralSym)
>     - Starting material guidance: 90% solve rate with 66% incorporating target materials (vs. 64% search-level guidance)
>     - 6.2 non-overlapping routes per target (3× more diverse than search-level methods)
> - **Constraint under multiple criteria (c )**: results for this application are underway.
>
> # Related work incomplete
>
> We thank the reviewer for pointing out this recent work. We have now included discussion of [3] in our revised manuscript in the following locations:
>  - Section 5 (Related Work, lines 466-480): We acknowledge LLM-guided search paradigms and discuss how these approaches operate by evaluating and re-ranking candidates from frozen single-step models based on natural language queries, fundamentally limiting exploration to chemical spaces accessible by the underlying model—the same limitation we address.
>  - Section 6 (Discussion and Limitations, lines 510-517): We explicitly discuss complementarity with LLM-based methods, noting that while search-level guidance (including LLM approaches) excels at interpreting natural language and re-ranking complete routes, our token-level guidance produces candidates inaccessible to unguided generators. We suggest integration where LLMs define constraints from natural language that our classifiers enforce during generation.
>
> A key difference with our method is that LLM approaches operate at the search level by selecting from pre-generated candidates (similar to other search-level methods like TangoStar), while we modify the generation process itself via token-level guidance to access previously unreachable chemical space
>
> # References
>
> [1] "FUDGE: Controlled Text Generation With Future Discriminators", Yang et al. 2021.
>
> [2] "Re-evaluating Retrosynthesis Algorithms with Syntheseus", Maziarz et al. 2023.
>
> [3] "Chemical reasoning in LLMs unlocks steerable synthesis planning and reaction mechanism elucidation", Bran et al. 2025

---

### Official Review · Reviewer_BgLs · 2025-11-04

**Soundness:** 1
**Presentation:** 1
**Contribution:** 2
**Rating:** 2
**Confidence:** 5

**Summary:**

In this paper, the authors develop a classifier guidance framework to modify the token-level logits of an autoregressive sequence-to-sequence model for retrosynthesis. The goal is to incorporate domain knowledge to retrosynthesis models. Evaluation is performed on USPTO-50k and USPTO-190, with the proposed rsmiles method.

**Strengths:**

* Incorporating domain knowledge into synthesis planning in retrosynthesis is an important and cutting-edge research topic in AI for Science.

**Weaknesses:**

* The method section does not include enough level of detail for readers to understand and reproduce your work. Specifically, the following important questions are not answered after reading your paper:
    * What are the building blocks of your proposed framework?
    * What is the algorithm framework? How does the proposed approach integrate into an existing sequence-to-sequence model?
    * What are the classifier models used? What are the architectures? How to train the classifiers?

  All details above should be clear to readers after reading your method section, instead of listing math equations.
* Figure 1 is hard to understand. The legend said "numbers represent the reaction types (0 to 11 classes)", but I cannot understand the meaning. I cannot tell the benefit of "steered" synthesis planning, either.
* Theorem 1 seems trivial, and I cannot understand the insight behind it. If, as discussed in L157, _the key insight is that token-level guidance accumulates exponentially across sequence length_, it seems an obvious intuition and does not need such a heavy theorem block. It will be much better if you save the space for methodological details.
* In the evaluation, only limited baselines are considered. I do not even find a place where the authors mention the name of the proposed approach. It is challenging to consider the contribution of this work if it does not consider the strongest baselines available (they are discussed in the introduction and related works). Also, an ablation study is missing.

**Questions:**

Please see weaknesses.

---

> ### Author Response · Authors · 2025-11-26
> **Response to Reviewer BgLs**
>
> We thank the reviewer for the opportunity to improve our work. We address the reviewer's concerns in turn:
>
> # Some clarifications
> - **"the authors develop a classifier guidance framework to modify the token-level logits of an autoregressive [...] model"**: we apply token-level classifier-guidance to a neural retrosynthesis model, **with additional results on its accessibility guarantees**. The novelty lies in our application domain, supplemented with a theoretical analysis on the necessity and ability of our token-level guidance when paired with beam search. We have rectified the introduction to reflect this distinction.
> - **"The goal is to incorporate domain knowledge to retrosynthesis models."**: while incorporating ground truth information is an important application of our method, **our broader goal is to allow practitioners to steer synthesis planning in any direction they desire, at inference time, by steering the output of the single-step model itself**. Previous approaches to this task focused on acting at the level of the search algorithm (through filtering single-step precursors [1], modifying the search policy to incorporate desired properties [2], or using LLMs to specify certain synthesis criteria in natural-language [3]). Our contribution towards this goal is to identify a gap in the guided synthesis planning literature: **the gains achieved from biasing the single-step model itself, in addition to guiding the search algorithm towards desirable components**.
>
> # The building blocks of our method & Integration into an existing model
> Our method works by training neural networks (classifiers or regressors depending on the property at hands) to predict the property from partially completed sequences, then we combine the logits of the predictor with those of a pre-trained generator.
> We made the following modifications to answer the reviewer's questions:
> - Section 2.2 now provides a complete introduction to classifier guidance for autoregressive models, including the mathematical formulation (Equations 1-4) and efficiency considerations,
> - Section 3.2 details our implementation: we use the pre-trained R-SMILES transformer as the base retrosynthesis model ($p_{\theta}$), keep its weights frozen, and combine its token predictions with a separately trained property classifier ($p_{\alpha}$) during beam search. This is illustrated in Figure 2,
> - Algorithm 1 (page 5) now provides the complete pseudocode showing exactly how classifier scores modify token logits at each generation step.
>
> # The classifier models used
>
> We implement these classifiers as Transformers following the same architectural design and data augmentation strategy as R-SMILES, but with a property prediction head and smaller network capacity (as specified in Section 3.2 (lines 212-218)). The architecture we use is a vanilla transformer encoder with 4 layers, 4 attention heads, 128 hidden dimensions, 0.1 dropout. We train the models with an Adam optimizer (lr=1e-3, weight decay=1e-5), for 2000 epochs in 12 hours on A100 GPU. The data used for training, as shown in Appendix B.2 with Figure 5, is made of partial sequences generated by truncating reactant SMILES at various completion ratios (minimum 0.8), concatenating with conditioning context (product for reaction type, starting material for Tanimoto similarity), and augmenting complete sequences using R-SMILES-style canonicalization (5 augmentations per complete sequence). We also show the performance of our classifiers through validation curves (Figure 6, Appendix C.1): we achieve 92% accuracy for reaction type on USPTO-190, R²=0.95 for Tanimoto similarity
>
> # Figure 1 is not clear
> We thank the reviewer for the opportunity to improve our figure. We have modified figure 1 to improve its clarity in the following ways:
> - We replaced numeric reaction type labels with distinct visual symbols (squares, triangles, circles) for immediate recognition, added a clear legend identifying "Chemist's preferred reaction types," and simplified the overall visual presentation.
> - The figure demonstrates that filtering (a) is overly restrictive and finds no routes. Search-level guidance (b) **can only work with reactions the single-step model already generates**, thus accepts a wrong reaction type at step 2, while our token-level guidance ( c) **modifies what the single-step model generates**, thus producing the correct reaction types at both steps and returning two routes meeting the chemist's specifications.
> - We have also added figure 2 (page 4), which provides a concrete molecular example showing token-level reranking during acetophenone synthesis, and figure 3b (page 7), which demonstrates chemical space exploration via Venn diagrams showing only 39% overlap between guided and unguided samples. These figures provide complementary evidence of our method's ability to access new regions of chemical space.

---

> > ### Author Response · Authors · 2025-11-26
> > **About Theorem 1**
> >
> > # Theorem 1 seems trivial
> > We appreciate the reviewer's feedback on Theorem 1. We have revised the presentation to better clarify the non-trivial contributions of this theoretical result. The reviewer is correct that exponential accumulation across sequence length is intuitive. However, the key insight of Theorem 1 is not the accumulation itself, but rather the **implications it has on the practical reachability under realistic computational constraints**
> > - Standard beam search prunes sequences based solely on base model probability $p_{\theta}(r \mid p)$. Property-satisfying sequences with low $p_{\theta}(r \mid p)$ are eliminated early and never reconsidered, regardless of how many beams we maintain. Guided beam search reranks tokens using both base model probability and property probability, allowing property-satisfying sequences to remain competitive in the beam despite lower $p_{\theta}(r \mid p)$.
> > - Theorem 1 formalizes this: it identifies conditions under which a specific property-satisfying sequence r* (that would be pruned by standard beam search with beam size k) can be recovered through guidance. The guidance scale $\lambda*$ acts as a "correction factor" that compensates for the base model's probability deficiency. The result hinges on the performance of the property predictors, which take in partial sequences, and therefore require enough generated context to make reasonable guesses.
> > - We now formalize the observation regarding the minimum length required for guidance to be significant in Corollary 1. The result establishes that guidance need only be applied after sufficient context accumulates (l ≥ l_min), reducing computational overhead while maintaining reachability guarantees. This directly motivated our algorithmic design choice of the minimum length threshold.

---

> ### Author Response · Authors · 2025-11-26
> **Additional results**
>
> # Additional Baselines
> We now compare against all retrosynthesis models available in Syntheseus [2], providing a comprehensive benchmark. This includes both template-based methods (LocalRetro, RetroKNN, NeuralSym) and template-free approaches (MHNReact, MEGAN, Graph2Edits, Chemformer, R-SMILES). We summarize our results in what follows, with additional details given in Sections 4.1 and 4.2:
> - **Single-Step Results (Table 1)**: Our token-guided R-SMILES achieves:
>     - 31% exact route recovery on USPTO-190-Steps (vs. 28% unguided R-SMILES, 16% NeuralSym)
>     - 42% recovery when guiding toward starting materials (vs. 31% for reaction type guidance)
>     - 23% correct reaction class (vs. 3% unguided), while maintaining 67% unique samples
>     - Critically, guided samples show only 39% overlap with unguided samples (Figure 3), demonstrating genuine exploration of new chemical space rather than reranking
> - **Recovery of Failed Cases**: Among 124 products where unguided R-SMILES fails, token guidance recovers 55% via round-trip accuracy and 10% via exact match (Figure 3), providing direct evidence that our approach accesses precursors inaccessible to the base model.
> - **Multi-Step Planning (Tables 2-3)**:
>     - 92% solve rate for reaction type guidance (vs. 81% unguided, 47% NeuralSym)
>     - 90% solve rate with 66% routes incorporating target starting materials (vs. 64% search-level guidance)
>     - 49% reactions match specified type (vs. 23% unguided)
>     - 6.2 non-overlapping routes per target (3× more diverse than search-level guidance).
>
> These results demonstrate that token-level guidance enables access to previously inaccessible chemical spaces during generation, translating to higher-quality multi-step synthesis plans that better satisfy chemist-specified constraints.
>
> # Ablations
>
> We thank the reviewer for the opportunity to strengthen our empirical results. We are working to provide the necessary ablations, which should be made available shortly.
>
> # References
>
> [1] "Planning chemical syntheses with deep neural networks and symbolic ai.," Segler et al., 2018
>
> [2] "Tango*: Constrained synthesis planning using chemically informed value functions", Armstrong et al., 2024
>
> [3] "Chemical reasoning in LLMs unlocks steerable synthesis planning and reaction mechanism elucidation", Bran et al., 2025.

---

### Author Response · Authors · 2025-11-26
**General response**

We thank the reviewers for their time and value feedback. Below we summarize the changes we made to our manuscript to address their concerns.

# Clarifying contribution w.r.t classifier-guidance framework

As reviewer tWiM pointed out, token-level guidance is indeed established in controllable text generation literature [1]. Our main contributions are: (1) formalizing accessibility guarantees through Theorem 1 and identifying key tuning parameters (guidance scale λ and minimum length threshold), (2) domain adaptation to retrosynthesis where molecular validity and reaction feasibility constraints differ fundamentally from text generation, and (3) demonstrating this as a viable approach for steering autoregressive SMILES-based retrosynthesis models. We have rectified the introduction (lines 38-48) to reflect these distinctions more clearly.

# Methodological details

We thank reviewer BgLs for prompting a more detailed presentation. Section 2.2 now provides complete mathematical formulation of classifier guidance for autoregressive models (Equations 1-4), Section 3.2 details our implementation with R-SMILES as the frozen base model, and Algorithm 1 (page 5) provides complete pseudocode. Figure 2 illustrates token-level reranking with a concrete molecular example. We have also simplified Theorem 1's presentation (lines 256-270) and added Corollary 1 establishing the minimum length threshold for effective guidance.

# Additional baselines

 Upon suggestions from reviewers tWiM, ZoTP, and nNwF, our experimental section now includes comparisons to all models available in Syntheseus [2]: template-free methods (MHNReact, MEGAN, Graph2Edits, Chemformer, R-SMILES) and template-based methods (LocalRetro, RetroKNN, NeuralSym). Results appear in Tables 1-3 and Sections 4.1-4.2. We also discuss complementarity with LLM-steered synthesis approaches [3] in Section 5 (lines 466-480) and Section 6 (lines 510-517).

# Ablations

We are working to provide ablations on: (1) single-step model guidance vs. search-level filtering, (2) guidance parameters (minimum length and λ selection), and (3) top-k token selection during guided generation. These will be made available shortly.

# General improvements to the empirical analysis

- Results now reported on entire USPTO-190 test set, with failure set analysis relegated to Figure 3a for comparison
- Multi-constraint experiments underway (combining reaction type and starting material guidance)
- Fair comparison framework: all baselines generate 100 samples with post-hoc filtering; our method uses 30-sample look-ahead for λ selection then generates 70 additional samples (100 total)
- Extended evaluation to USPTO-50k (single-step) and Pistachio (multi-step) in progress

# References
[1] "FUDGE: Controlled Text Generation With Future Discriminators", Yang et al., 2021.

[2] "Re-evaluating Retrosynthesis Algorithms with Syntheseus", Maziarz et al., 2023.

[3] "Chemical reasoning in LLMs unlocks steerable synthesis planning",  Bran et al. 2025.

---

### Author Response · Authors · 2025-12-04
**Thank you for the valuable feedback. Additional experiments will be included in future revisions.**

We thank the reviewers for their time and thoughtful feedback. The suggestions regarding ablation studies (guidance scale, minimum length, number of guided tokens), evaluation on additional datasets (USPTO-50k, Pistachio), and combining multiple properties have been valuable for identifying directions to strengthen the work. We will continue addressing these points in future revisions.

---

### Meta-Review · Area_Chair_y272 · 2026-01-08

**Summary:**

This paper proposes token-level classifier-guided decoding for steerable retrosynthesis planning, aiming to bias a frozen single-step model toward user-specified constraints without retraining. Reviewers found the problem setting relevant and the idea of guiding generation (rather than post-hoc search) interesting. However, concerns were raised regarding overstated novelty relative to prior classifier-guidance work, limited practical justification of the steering scenarios, and insufficiently convincing empirical evidence for the claim of accessing fundamentally “inaccessible” chemical space. Despite substantial clarifications and additional experiments in the rebuttal, the reviewers remained unconvinced that the current submission can be accepted.

**Reviewer Concerns:**

The authors addressed several clarity issues in the method description, added additional baselines, and improved positioning relative to prior work. However, key concerns remain unresolved. Reviewers continued to question the incremental nature of the contribution, the realism and fairness of the evaluation setup (e.g., reliance on ground-truth or pseudo-label information, product-specific test-time parameter selection), and the practical usefulness of the proposed steering mechanisms. Experimental validation is still largely limited to USPTO-style datasets, and stronger evidence is needed to support the central claims.

**Reviewer Scores:**

All reviewers maintained rejection recommendations. While the rebuttal improved clarity and completeness, no reviewer indicated a score change that would materially affect the final decision.

---

### Decision · Program_Chairs · 2026-01-26

Reject